



# Hyper-resolution flood hazard mapping at the national scale

Günter Blöschl[a], Andreas Buttinger-Kreuzhuber[ac], Daniel Cornel[c], Julia Eisl[d], Michael Hofer[d], Markus Hollaus[b], Zsolt Horváth[ac], Jürgen Komma[a], Artem Konev[c], Juraj Parajka[a], Norbert Pfeifer[b], Andreas Reithofer[d], José Salinas[a], Peter Valent[a], Roman Výleta[e], Jürgen Waser[c], Michael H. Wimmer[b], Heinz Stiefelmeyer[f]

[a] TU Wien, Institute of Hydraulic Engineering and Water Resources Management, Karlsplatz 13, 1040 Wien, Austria

[b] TU Wien, Department of Geodesy and Geoinformation, Research Area Photogrammetry, Wiedner Hauptstraße 8-10, 1040 Wien, Austria

[c] VRVis Zentrum für Virtual Reality und Visualisierung Forschungs-GmbH, Donau-City-Strasse 11, 1220 Wien, Austria

[d] Ingenieurbüro Dipl.- Ing. Günter Humer GmbH - Ingenieurbüro für Kulturtechnik und Wasserwirtschaft, Geboltskirchen/Gmunden, Austria

[e] STU Bratislava, Faculty of Civil Engineering, Department of Land and Water Resources Management, Radlinského 11, 810 05 Bratislava, Slovakia

[f] Federal Ministry of Agriculture, Forestry, Regions and Water Management, Stubenring 1, 1010 Wien, Austria

*Correspondence to:* Günter Blöschl (bloeschl@hydro.tuwien.ac.at)

**Abstract.** Flood hazard mapping is currently in a transitional phase involving the use of data and methods that were traditionally the domain of local studies in a regional or nation-wide context. Challenges include the representation of local information such as hydrological particularities and small hydraulic structures, and computational and labour costs. This paper proposes a methodology of flood hazard mapping that merges the best of the two worlds (local and regional studies) based on experiences in Austria. The analysis steps include (a) quality control and correction of river network and catchment boundary data; (b) estimation of flood discharge peaks and volumes on the entire river network; (c) creation of a digital elevation model (DEM) that is consistent with all relevant flood information, including river bed geometry; (d) simulation of inundation patterns and velocities associated with a consistent flood return period along the entire river network. In each step, automatic methods are combined with manual interventions in order to maximise the efficiency and at the same time ensure estimation accuracy similar to that of local studies. The accuracy of the estimates is evaluated in each step. The study uses flood discharge records from 781 stations to estimate flood hazard patterns of a given return period at a resolution of 2 m over a total stream length of 38000 km. It is argued that a combined local-regional methodology will advance flood mapping, making it even more useful in nation-wide or global contexts.



## 1 Introduction

Flood hazard mapping is a key tool in integrated flood risk management, given that numerous management measures hinge on the knowledge of the inundation probability of different parts of the landscape. For example, structural flood defence measures, such as levees and retention basins all need to be designed to specific probabilities (e.g. an exceedance probability of 1 in 100 years, equivalent to a return period of 100 yrs) in order to balance economic and safety considerations. Similarly, non-structural measures such as evacuation plans and insurance coverage rely directly on accurate estimates of inundation probability (FEH, 1999).

The traditional approach to flood hazard mapping is based on local experience. Typically, field reconnaissance is combined with local experience from past floods to delineate areas in the landscape of similar probability of flooding (Díez-Herrero et al., 2009; Mudashiru et al., 2021). While theoretically not very rigorous, the resulting maps are usually very reliable because they account for the local particularities of the river reach including local walls, culverts, and other hydraulic structures that affect the inundation behaviour of the stream locally. Over the years, these approaches have been complemented by local hydrodynamic simulations over a few stream kilometres that allow for a more objective mapping of flood areas with different probabilities, and in which the local particularities are incorporated, using the same field information (e.g. Syme and McColm, 1990). Sometimes, a single opening in the system may completely change the inundation behaviour, e.g. as in the case of the 2005 flood in Mittersill, Austria, where a single railway underpass left open resulted in the flooding of half the township. Local particularities may therefore be essential for accurate flood hazard mapping.

More recent developments in simulation methods and data availability have opened the potential of larger scale simulations which respond well to new needs for large scale hazard maps in strategic risk assessments (Sayers, 2013; Alfieri et al., 2014; Ward et al., 2015). The increased simulation capabilities are mainly related to increased computational power, e.g. brought about by parallel computing, and to some degree to more efficient methods, such as solvers of the underlying equations and associated numerical implementations (e.g. Buttinger-Kreuzhuber, 2019; Horváth et al., 2020). For example, while in the 1990s typical hydrodynamic simulation problems involved thousands of simulation cells, today a billion cells are no exception (Jankowfsky et al., 2016; Assteerawatt et al., 2016; Hoch and Trigg, 2019).

The increased data availability is in line with a general trend towards digitalisation. Again, in the 1990s digital elevation models were often based on conventional topographic surveys, while today Lidar based elevation models feature resolutions of less than a meter, accuracies of a few centimetres and they are often available at a national scale (Pfeifer and Mandlburger, 2017). These data are complemented by numerous other geo-datasets, some of them open, such as open street maps and land use data as well as satellite based inundation patterns for model calibration (Domeneghetti et al., 2019).

While this trend towards larger scale flood hazard mapping is continuing, there are a number of challenges when moving from local to regional or national scales. One challenge is related to the extent to which local particularities of the hydrological and hydrodynamic system can be captured at large scales. This is because high resolution does not necessarily imply high accuracy, unless local information is included (Savage et al., 2016; Trigg et al., 2016). Fine scale detail on the



effect of cracking soils and preferential flow on infiltration of rainwater into the soil, for example, may very much matter for

estimating flood peak magnitudes at larger scales, as highlighted by the ungauged catchment problem (Blöschl et al., 2013). Similarly, in estimating overland and river flow it is not possible to fully capture all the detail of each kerb and culvert at regional scale. There will therefore always be some level of generalisation or simplification. The quality of the hazard maps produced thus hinges on the appropriate way of generalisation and the degree to which heterogeneous data can be combined at the required scale.

A related challenge is the practical work flow. In local studies, manual interventions in the work flow can be very efficient because of their flexibility, and the analyst can inject all their knowledge about the flood processes in the particular area. At large scale automatic methods are the norm, and manual interventions may become prohibitively expensive simply because of labour costs. The question then remains how local experience and knowledge can be accounted for.

A third challenge when moving from local to larger scales is related to the simulation of inundation patterns of a given

exceedance probability or return period. Local studies usually focus on one river reach, trying to mimic an extreme flood, similar to one that has happened or could happen in the future (e.g. Horváth et al., 2020). The simulations thus use, as an upper boundary condition to the hydrodynamic simulations, a streamflow hydrograph associated with a peak of the required return period. The underlying concept of a scenario is however no longer valid if a region is considered instead of a river reach, as floods never exhibit the same return period over large regions. In fact, their spatial variations are the defining

characteristics that make specific flood discharge of a given return period almost always decrease downstream. Because of this, it is no longer possible to adopt a scenario approach to inundation simulations for large regions, and alternative concepts are needed.

The aim of this paper is to propose a framework of hyper-resolution fluvial flood hazard estimation that addresses all three challenges. It combines representations of local information, such as hydrological particularities and small hydraulic

structures, with large scale data and simulations, keeping in mind the computation and labour costs at that scale. The paper goes beyond existing studies in that a greater emphasis is on the exploiting of local information through combined automatic-manual methods. It proposes as new method of simulating inundation maps with constant return periods along the entire river network. In the national context, these maps serve three purposes: The Austrian Association of Insurance Companies (VVO) and their members use them as a tool for premium estimation as floods have recently become insurable in

Austria; the Austrian Federal Ministry of Agriculture, Tourism and Regions uses them to comply with the flood risk zoning requirements of the EU flood directive (EU 2006); and they are used more generally in the public sector of natural hazard communication to enhances risk awareness among the citizenship.



## 2. Overall framework

The overall conceptualisation of the workflow of the mapping has been gleaned from previous studies (e.g. Merz et al., 2008), however, with a greater emphasis on exploiting local information and a view of spatially consistent return periods. There are four main datasets that connect the various process parts and the collaborators.

(a) The first is the overall tessellation of the landscape into vector data of a river network and associated catchment boundaries (indicated in pink in Figure 1). The purpose is to provide the backbone of the fluvial flow processes in the

landscape, which is used to structure the remaining analyses.

(b) The second are the flood discharge estimates (indicated in green in Figure 1) on the entire river network. In this study, we chose a regional flood frequency approach rather than a rainfall runoff modelling approach as is sometimes adopted in similar studies. The choice was motivated by the usually large biases of the derived flood frequency approach that estimates flood probabilities from precipitation, and the availability of numerous long discharge records in the study region. Indeed,

practical flood frequency estimation almost always hinges on observed flood series, if available, because of their potential in bias reduction (Rogger et al., 20xx). The choice of the regional flood frequency approach also explains the need for vector data of a river network with correct topology (upstream, downstream, confluences), and stream gauges linked to the river network, in order to regionalise the flood discharges estimated for the stream gauges to the entire river network.

(c) The third set of information revolves about a geometric representation of the landscape in terms of digital elevation

models (DEMs) and landscape descriptors (indicated in orange in Figure 1). These are the basis of the hydrodynamic simulations and link them with the flood discharge information along the river network of dataset (b). Given its role as a connective tissue in the framework, its precise consistency with the other data is essential. The DEM is based on Lidar data with a resolution of less than a meter, modified to enable the free flow downstream.

(d) The fourth piece of information represents the water flow in and near the river system, specifically the inundation

patterns, depths of inundation and the flow velocities (indicated in blue in Figure 1). We solve the transient 2D shallow water equations using flood discharge hydrographs as inputs along the (vector) river network, and account for hydraulic structures and other local particularities, assimilating heterogeneous information, including proxies for surface roughness. The simulations have been set up in a way to directly obtain inundation patterns and velocities associated with a consistent flood return period along the entire river network, as opposed to event-based simulations.

These four main datasets are linked through exchange of information as indicated by the arrows in Figure 1. In reality, the interaction of the components of Figure 1 is more complex, and in many instances there, were iterative and parallel interactions in different parts of the study regions. In line with the main aim of the paper, the emphasis was on exploiting local information through combined automatic-manual methods for which no off-the-shelf methods were available, and thus suitable procedures had to be developed to account for the particular hydrological and/or data situation. Many of them

involved a combination of automatic methods and manual interventions by analysts. The purpose of the former was to speed up the processes given the large number of nodes or cells in the system (a total of 19479 nodes of the vector river network



and 20 billion simulation cells of the raster data set), while the purpose of the latter was to deal in a more accurate way with those situations where the automated method failed because of local particularities. The general approach to this automatic-manual combination was the following:

- Developing an automatic method (or using one from existing studies);
- Testing the method against more detailed data (often in a small part of the study region);
- Finding criteria for the applicability of the automatic method; and
- Dealing with the remaining cases manually.

In this way we strive to combine the best of two worlds: low labour costs of the analysts as only a small fraction of the
nodes/cells requires manual intervention; and high accuracy as the approach allows capturing complex hydrological and hydrodynamic situations.

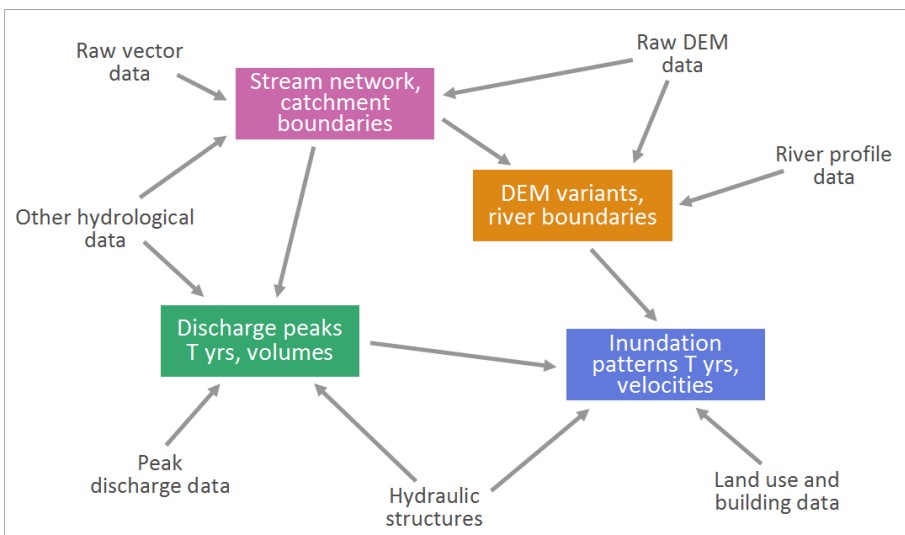

**Figure 1: Main datasets (text) and information flows (arrows) of the flood hazard mapping approach presented here. "T yrs"**
**refers to the return period of the discharge peaks and the inundation patterns.**

## 3. River network and catchment boundaries

### 3.1 Data and methods used

The available dataset of catchment boundaries contained 41069 polygons, each of which represented a small sub-catchment
area with a mean size of 1.5 km². The data also contained information on the next downstream sub-catchment polygon, enabling construction of a complete topology of the Austrian catchments without the need to derive this information from the





DEM. In 390 cases, the runoff from one sub-catchment was divided into more than one downstream sub-catchment because of diversion canals, e.g. for historic mills, hydropower, irrigation and flood protection. As the method of flood peak regionalisation (see section 4.1) requires a stream topology without bifurcations, the cases had to be identified and the

bifurcations removed.

As the interest of the present study was on streams draining catchments greater 10 km², streams with smaller catchment areas were removed automatically, which resulted in a total river network length of about 38 000 km. The river network dataset also contained numerous canals of different types (see red lines in Figure 2), which were not distinguishable from the natural rivers based on the information in the dataset. Manual identification of the canals was therefore needed. Figure 2b shows an

example, where an irrigation canal diverts water from the Große Tulln River to flow through the nearby villages. Figure 2c shows an example upstream of St. Pölten, where part of the Traisen's discharge is diverted into a network of canals for small hydro power. In both examples, the main river was retained and the canals were removed manually.

In the available dataset, the river network and the catchment boundaries were not consistent because of the different origins of the data. Since the regionalisation of the flood discharges to the river network does require consistency, it had to be

established by intersecting the datasets. In this process the dataset of river network was divided into a number of sections each of them assigned to one polygon of the catchment boundaries dataset. The downslope connectivity and a unique catchment-river alignment were automatically checked and particular cases were corrected manually against the DEM and aerial photographs.

Finally, the catchment boundaries and river network datasets were supplemented by a node dataset of river profiles (see red

points in the lower row in Figure 3) that were generated at both ends of each river segment. The nodes represent outlets from the catchments for which all the flood characteristics including the T-year floods were estimated.

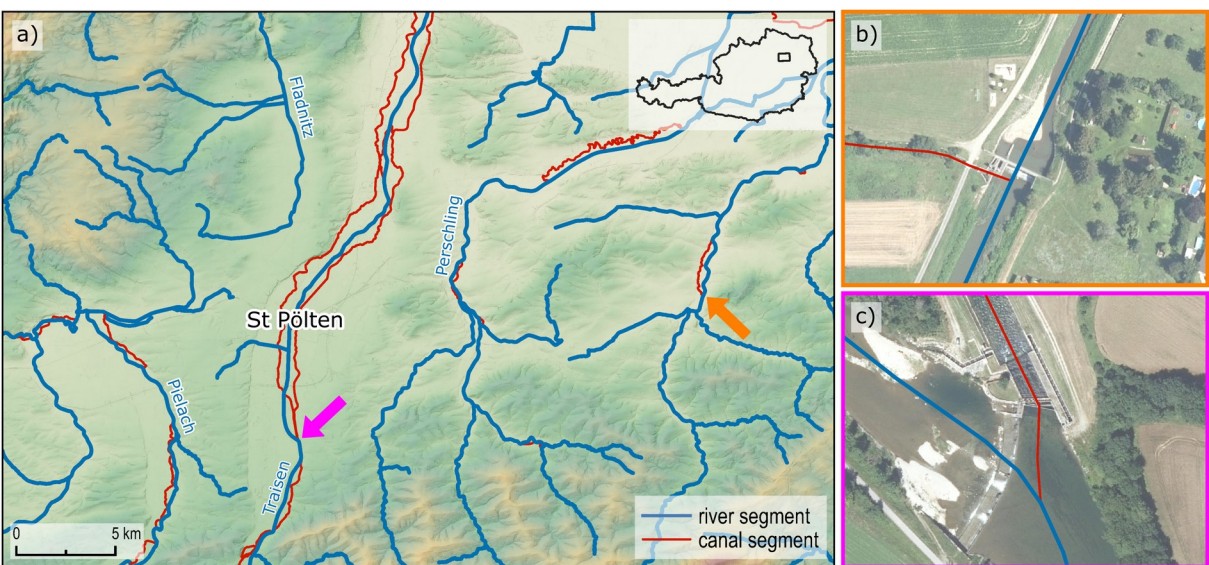

**Figure 2: (a) Generalisation of the river network (blue) by removing canals (red) in part of Lower Austria. (b) Irrigation canal diverting water from the Große Tulln. (c) Canal diverting water from the Traisen, formerly used for mills and now for hydropower (e.g. the Luggau plant).**

## 3.2 In what way the combined automatic-manual methods work

In a first step, the original sub-catchment dataset was checked for consistency by automatically identifying headwater catchments, sub-catchments, water diversions and cumulative catchment areas. The removal of the 390 channels required a decision on which branch to keep, which was made on the basis of a hydrological interpretation of the orthophotos. In most instances, the natural river was retained, but in those cases where the canal was for flood mitigation purposes that canal was retained.

In order to link the river network with their catchments, the river network was intersected with the sub-catchment boundaries, which resulted in the creation of nodes on the river network. Because of position inaccuracies, occasionally more than one river section was assigned to a sub-catchment. These cases were identified automatically and dealt with manually, resulting in either a merger of the affected river segments or in their association with neighbouring sub-catchments. In the example of Figure 3 left, a reach of the Deutsche Thaya River intersects the boundary of its tributary. The manual intervention consisted of merging the two outer nodes with that of the confluence. Figure 3 right shows an off-level crossing of the Frutz and Ehbach Rivers. While the automatic procedure associated the culvert reach of the Ehbach to the catchment of the Frutz River, this association was removed by the manual intervention.



### 3.3 Accuracy of the results

A combination of automatic and manual checks was deemed to be most efficient for assessing the accuracy of the topology of the combined river network - catchment boundary dataset. The automated checks included testing the monotonicity of catchment size along the river network, verifying the calculated catchment sizes of stream gauges against those published in the hydrological year books, and checking the consistency of neighbouring upstream and downstream river sections and catchments. Any inconsistencies identified were dealt with manually. Additionally, the entire dataset was checked visually

against orthophotos and the digital terrain model, including the linkage between the river reaches and their catchments (Figure 3). To facilitate the comparison, matching colours were used.

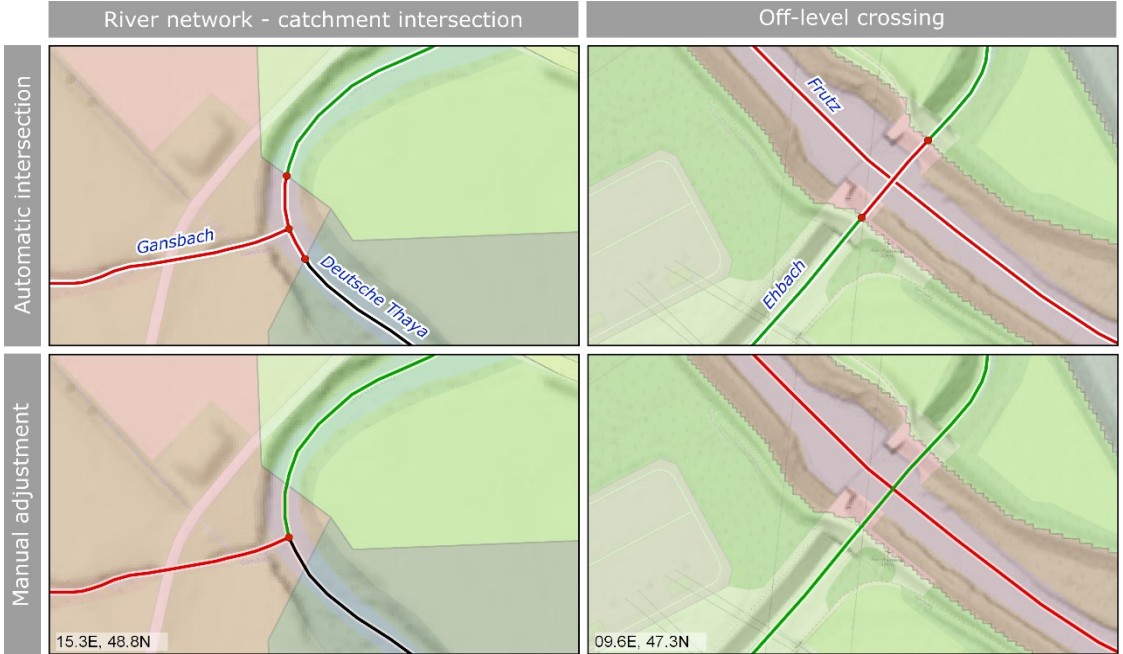

**Figure 3: Automatic (upper row) and manual (lower row) procedures to associate the catchment boundaries with the river**

**network. Left: Merging of the two outer red nodes that arise from position accuracies of the catchment boundary of the Gansbach, tributary to the Deutsche Thaya, Lower Austria. Right: Ehbach crossing under the Frutz in Vorarlberg. The river reach and associated catchment area have matching colours.**



## 4. Flood discharge peaks and volumes

### 4.1 Data and methods used

(a) Flood data quality control

The flood data used consist of time series of maximum annual flood peaks for 781 catchments. The size of the catchments ranged between 1.6 and 131,488 km² with 25% of the catchments being smaller than 55 km² and 75% smaller than 382 km². In most catchments, the flood peak records were available until 2019. Record lengths ranged between 5 and 192 years (Donau at Linz) with a median of 40 years (Figure 4). Most of the time series were provided by the Hydrographic Services

of Austria, some by hydropower companies and some were obtained from neighbouring countries. The flood data were thoroughly examined, starting with checking the location of the stations and its association with the river network and catchment boundary. The flood series were visually inspected and in case of any irregularities, the responsible staff member of the Hydrographic Services was interviewed (Merz et al., 2008). The interviews usually revolved around outliers, which were sometimes caused by ice jams or wooden debris jams. In such cases the records were corrected. Some records were

affected by hydraulic structures or channel regulation. In such cases the flood records were split into a pre-change and post-change part. The post-change parts was considered to be locally relevant (downstream of the station), while the pre-change part of the record was considered regionally relevant (upstream of the station and in neighbouring catchments).

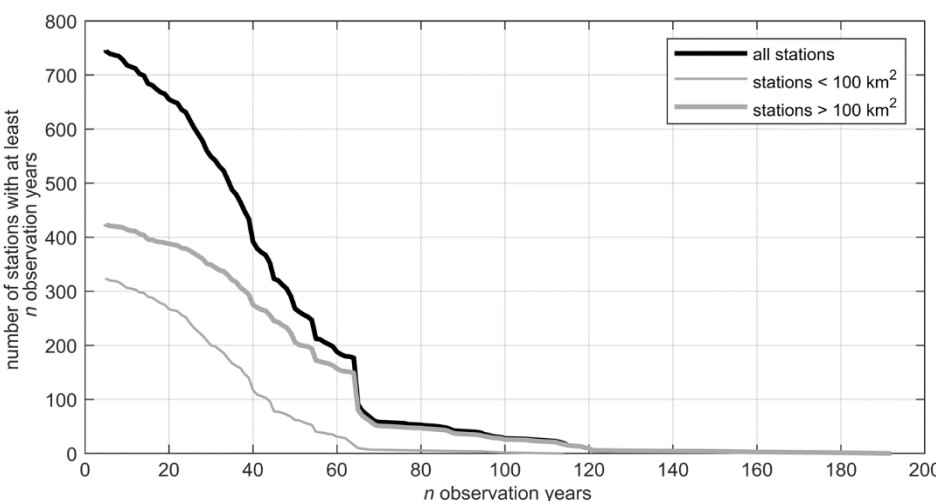

**Figure 4: Cumulative distribution of flood record length of the 781 stream gauges in Austria used here (thick black line), and**
**corresponding distributions for catchments <100 km² and >100 km² (thin and thick grey lines, respectively).**

(b) Local flood peak estimation for gauged basins

The pre-processed flood series from 781 stations were used to estimate the first three statistical product moments: mean annual specific flood ($MAF$), coefficient of variation ($C_V$) and coefficient of skewness ($C_S$). From these moments the T-year





flood discharges were estimated assuming a Generalized Extreme Value (GEV) distribution, as this was shown to be a
suitable methodological choice for Austria (Merz and Blöschl, 2005). However, since the estimation of T-year floods on the
basis of the flood series alone tends to be uncertain due to measurement errors, short record lengths and non-stationarities,
the flood frequency hydrology approach was adopted, in which additional information is used to improve the estimates
accuracy (Merz and Blöschl, 2008a, 2008b; Viglione et al., 2013). Three types of information expansion, encompassing both

quantitative and qualitative (soft) information, were considered. Temporal information expansion comprised longer records
from neighbouring stations to account for climate fluctuations, and historical floods to better constrain $C_S$. Spatial
information expansion consisted of comparisons with other gauged catchments in the regions accounting for differences in
geology and rainfall, using maps and discharge-area plots for visualisation. Causal information expansion involved
consideration of flood generation processes, e.g. using rainfall frequency plots within the Gradex method (Guillot, 1972) and

orthophotos and soil maps for assessing infiltration characteristics. Using this additional information, the statistical moments
estimated directly from the series were modified for about one third of the catchments.

(c) Regional flood peak estimation for ungauged basins

In the remaining nearly 21,000 ungauged basins the T-year flood discharges were also estimated from the three statistical
moments assuming a GEV distribution. The statistical moments were estimated by Top-kriging (Skøien el al., 2006), which

performs well as compared to other regionalisation methods (Merz and Blöschl, 2005; Salinas et al., 2013; Persiano et al.,
2021). It is based on spatial correlations, accounting for connectivity along the river network and local estimation
uncertainty, so that short series can be used with profit. In order to make allowance for controls other than spatial distance,
$MAF$ and $C_V$ were transformed prior to Top-kriging:

$$MAF^* = \ln\left(MAF \cdot A^{-1+\beta} \cdot \alpha^{-\beta} \cdot FARL^{-\gamma}\right) \quad MAF^* = \ln\left(MAF \cdot A^{\beta} \cdot \alpha^{-\beta} \cdot FARL^{-\gamma}\right) \quad \text{Eq. 1}$$

where $A$ is catchment area, $\alpha$ a reference catchment area of 100 km² and $\beta$ the slope of the relationship between $MAF$ and $A$
in a logarithmic scale. The logarithmic transformation reduces the skewness of $MAF$. $\beta$ depends on the flood process type
with values around 0.25 in regions where synoptic events or snow melt are relevant, and around 0.4 where convective storms
are relevant, reflecting the stronger spatial decorrelation in the latter case (Merz and Blöschl, 2009). The impact of the lakes
and reservoirs with permanent water was accounted for using the Flood Attenuation by Reservoirs and Lakes index FARL

(IH, 1999). The index was calculated for each catchment and can take values between 0 and 1 (1 for no lakes and reservoirs).
The scaling parameter $\gamma$ was set to 1.5 (Merz et al., 2008). Previous studies have identified a strong correlation between the
values of mean annual precipitation ($MAP$) and $MAF$ in Austria (Lun et al., 2021; Merz and Blöschl, 2005, 2009). In order to
enhance the robustness of the estimate, the estimated MAF in catchments smaller than 200 km² were adjusted in a post-
processing procedure using a regional regression between $MAP$ and specific $MAF$. The degree of adjustment was assumed to

be proportional to the estimated Top-kriging variance.

Merz and Blöschl (2009) showed that, in Austria, $CV$ decreases with catchment area more strongly for small catchments than
for larger catchments, so $C_V$ was normalized to a reference area of $\alpha = 100$ km² as:




$$C_V{}^\bullet = \begin{cases} C_V \cdot A^\beta \cdot \alpha^{-\beta}, & A < 100\ km^2 \\ C_V, & A \geq 100\ km^2 \end{cases} \qquad C_V{}^\bullet = \begin{cases} C_V \cdot A^\beta \cdot \alpha^{-\beta}, & A < 100\ km^2 \\ C_V, & A \geq 100\ km^2 \end{cases} \qquad \text{Eq. 2}$$

where $\beta$ was set to 0.1 based on an analysis of the data.

The third moment $C_S$ was regionalized without any transformation. The final step of the automatic procedure was the back-transformation of the moments after regionalisation from the T-year peak discharges were estimated for all 21,000 basins (Figure 5a).

(d) Shape of the flood hydrograph

For solving the transient 2D shallow water equations in the inundation modelling, discharge hydrographs are prescribed at all nodes so, in addition to the peaks, the shape of the hydrographs were estimated. It was assumed that the shape conforms to a gamma distribution with two parameters, peak and a time scale $T_C$. The latter represents the ratio of volume and peak discharge. $T_C$ values obtained by Gaál et al. (2012) from data of 396 catchments were scaled by catchment area to account for larger $T_C$ in bigger catchments, logarithmically transformed, regionalised by Top-kriging and back transformed / back

scaled. The resulting pattern is shown in Figure 5b. $T_C$ values range from several hours in small catchments, in particularly in the flash flood region in the South-east of the country, to around 100 hours for the Danube.

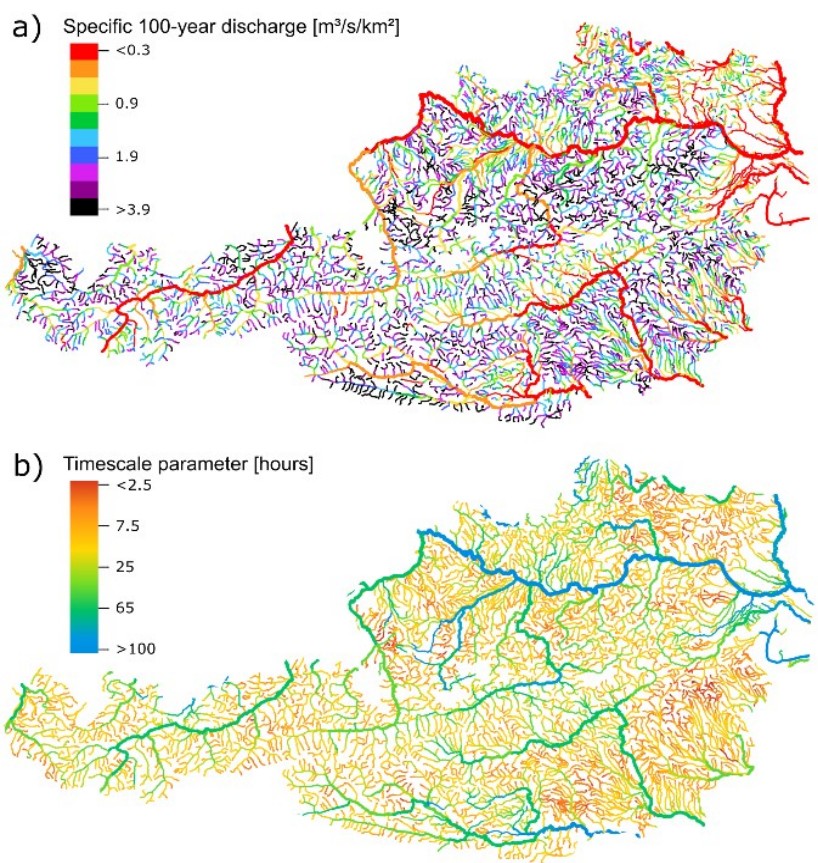

**Figure 5: (a) Specific 100-year flood discharge, (b) Timescale parameter $T_C$ specifying the duration of the flood hydrographs considered relevant. Both estimates have been regionalised to the entire river network of Austria using Top-kriging and are used as inputs to the hydrodynamic modelling.**

(e) Retention basins

Numerous dry flood retention basins exist in Austria, in particular in the South-East, out of which 398 have a storage volume of more than 10,000 m³. These basins lack permanent water bodies, so they were not accounted for by the FARL index. An alternative method was therefore developed in which the inflow flood hydrograph was transformed through a representative retention basin as a surrogate of all the retention basins upstream the catchment outlet. The retention basins were characterized by their storage volume ($V$), percent contributing area ($A_P$) and their arrangement (in series or in parallel). For reservoirs in series, the representative basin's volume was assumed to be the sum of the individual volumes, and its contributing area was assumed to be the weighted sum (using $V$ as the weight) of the affected upstream basins. For reservoirs in parallel, the approach was reversed ($A_P$ as a sum and $V$ as a weighted sum using $A_P$ as the weight).

In each catchment, only the portion of the flood hydrograph corresponding to $A_P$ was subject to flood peak reduction. The reduction was estimated by aligning the storage volume of the representative retention basin with the volume between the

inflow flood hydrograph and the line connecting the 2-year discharge, representing the discharge at which the retention basin

starts filling, and the reduced peak discharge (located on the falling limb). The method is automatic and was checked manually. The percent contributing area quickly diminishes along the river network and so does the effect of all the upstream reservoirs on reducing the flood peak discharges (Figure 6).

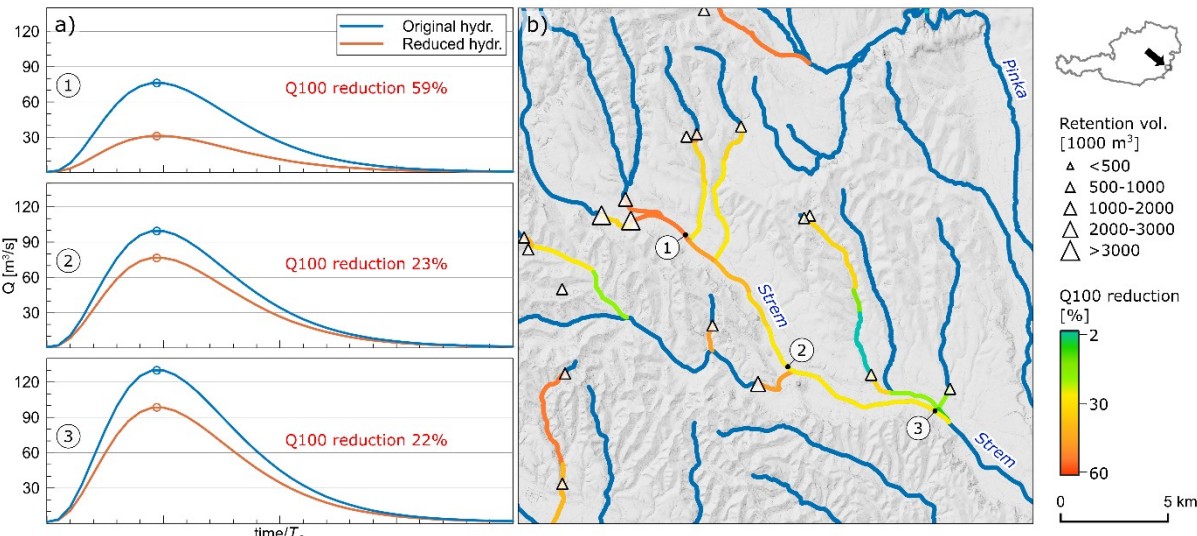

**Figure 6: Effect of retention basins on floods along the river network. (a) Original flood hydrograph (blue) constructed using the**
**regionalised statistical moments and reduced flood hydrograph (orange). Circles indicate the 100-year discharges $Q_{100}$. (b) The**
**efficiency of the retention basins in reducing $Q_{100}$ diminishes along the river network. Strem and tributaries, Burgenland.**

## 4.2 In what way the combined automatic-manual methods work

The combined, iterative approach of estimating flood discharge peaks and volumes leverages the strengths of automatic
methods at the regional scale while incorporating manual adjustments to address local peculiarities through auxiliary information. The flood data were automatically screened for outliers, and manually checked for any errors. Next, the flood moments were automatically estimated from the series and manually adjusted by expert judgement on the basis of auxiliary information, and re-evaluated after a personal discussion with staff members of the Hydrographic Services. In a next step, the estimated flood moments were automatically regionalized to the ungauged catchments, and the effect of retention basins
was automatically accounted for. The resulting modifications were visually checked through maps, and evaluated by expert judgement. The timescale parameters of the hydrographs were regionalized automatically and visually checked using maps.

*4.3 Accuracy of the results*





The predictive performance of the automatic regionalization method to ungauged catchments was evaluated by leave-one-out cross-validation for the locations of the stream gauges. The 100-year flood discharges $Q_{100,i}^{reg}$ $Q_{100,i}^{reg}$ were estimated via the

regionalised flood moments for which only the regional information was used and subsequently compared with the local estimates $Q_{100,i}^{loc}$ $Q_{100,i}^{loc}$ from the observed flood series. The predictive performance was evaluated by the relative bias, *RBIAS*, and the root mean squared normalised error, *RMSNE*:

$$RBIAS = \frac{1}{N}\sum_{i=1}^{N}\left(\frac{Q_{100,i}^{reg} - Q_{100,i}^{loc}}{Q_{100,i}^{loc}}\right) RBIAS = \frac{1}{N}\sum_{i=1}^{N}\left(\frac{Q_{100,i}^{reg} - Q_{100,i}^{loc}}{Q_{100,i}^{loc}}\right)$$ 

Eq. 3

$$RMSNE = \sqrt{\frac{1}{N}\sum_{i=1}^{N}\left(\frac{Q_{100,i}^{reg} - Q_{100,i}^{loc}}{Q_{100,i}^{loc}}\right)^2} RMSNE = \sqrt{\frac{1}{N}\sum_{i=1}^{N}\left(\frac{Q_{100,i}^{reg} - Q_{100,i}^{loc}}{Q_{100,i}^{loc}}\right)^2}$$

Eq. 4

where $N = 781$ is the total number of stream gauges, and *i* refers to a particular gauge.

The cross-validation was carried out for three regionalisation methods: ordinary kriging (OK) using catchment centroids to calculate distances between the catchments, Top-kriging (TK1) without considering, reservoirs, lakes and mean annual precipitation, and Top-kriging (TK2) accounting for these factors. Ordinary kriging gives the largest biases (RBIAS) and errors (RMSNE) (Table 1), as it relies solely on distance between catchments, potentially assigning more weight to gauges

outside the catchment of interest than to upstream and downstream neighbours. Top-kriging (TK1) addressed this issue, significantly reducing the percentage errors, especially for the larger catchments, and further improvements are achieved when accounting for reservoirs, lakes and precipitation (TK2). Biases are 6% for the smallest catchments and 1% for medium ad large catchments. The RMSNE of TK2 compare well with values found in the literature. RMSNE typically ranges around 0.4 – 0.5 in humid climates and is larger in tropical and arid climates (see Figures 9.25-9.27 of Blöschl et al.,

2013). The decrease of bias and error with catchment size is related to neighbouring gauges being more informative for larger catchments, again consistent with the literature (see e.g. Figures 9.28 and 9.29 of Blöschl et al., 2013; Persiano et al. (2021). An example of the comparison is shown in Figure 7. For the lowest reach of the Leising, a tributary to the Mur river, ordinary kriging gives a 100-year flood of 91 m³/s while Top-kriging gives 58 m³/s. The latter value is clearly more accurate, and reflects the ability of Top-kriging to account for the river network topology.






**Table 1: Prediction performance of the 100-year flood discharge estimates based on leave-one-out cross-validation for three regionalization methods and three catchment area ranges. OK - ordinary kriging, TK1 - Top-kriging without considering reservoirs, lakes and mean annual precipitation, TK2 - Top-kriging considering the above.**

| Catchment area [km$^2$] | Number of catchments | RBIAS \| RMSNE | | |
|---|---|---|---|---|
| | | OK | TK1 | TK2 |
| <100 | 312 | +0.125 \| 0.728 | +0.116 \| 0.558 | +0.063 \| 0.301 |
| 100-1000 | 375 | +0.191 \| 1.153 | +0.079 \| 0.550 | +0.008 \| 0.298 |
| >1000 | 94 | +0.041 \| 0.255 | −0.006 \| 0.067 | −0.010 \| 0.064 |

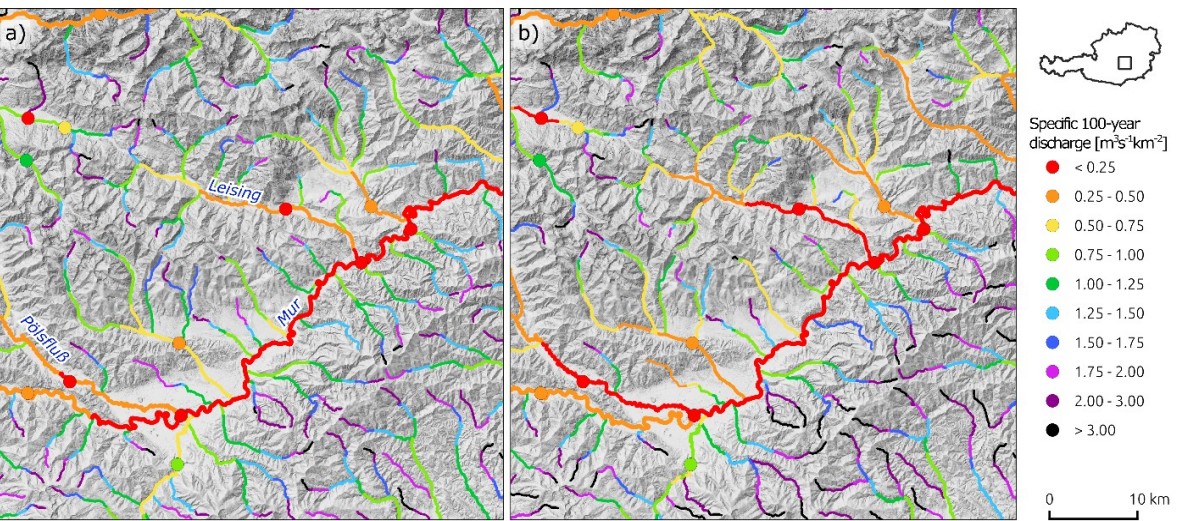


**Figure 7: Estimated 100-year specific flood discharges in the Upper Mur catchment near Zeltweg, Styria. Circles indicate the estimates at the stream gauges, lines the estimates on the river network, i.e. the ungauged catchments. a) Ordinary kriging without any pre- and post-processing, b) Top-kriging with pre- and post-processing.**

**5. DEM variants and river boundaries**

**5.1 Data and methods used**

The digital elevation model (DEM) is based on LiDAR flights from 2006-2014 with a spatial resolution of 1 m. Additionally, a digital surface model (1 m) derived from the LiDAR data and a digital orthophoto (DOP, 2013-2015, 0.2 m) including meta information were used. The relevant offices of the nine Austrian states had processed the data in slightly





345 different ways, resulting in some heterogeneities at the state borders, which were corrected by a filtering procedure (Wimmer et al., 2021).

The original vector dataset of the river network (i.e. the river axes) obtained from the relevant federal ministry showed some offsets from the river channel representation of the DEM due to time differences of data collection and generalisation in the digitalisation of the river axes. As consistency of these two datasets was deemed essential for all further data processing, an

350 automatic, cross-section-based procedure was implemented that sequentially corrects the river axis positions using the following criteria in descending order of priority: minimum height increases along the river axis; water passing through the deepest point of the river bed in a cross-section of the DEM; small deviation from the initial flow axis. All conditions were formulated as weight functions and combined for each cross section to find the most likely river course according to the DEM. An example is shown in Figure 8.

355 In a next step, bridges and other obstacles such as vegetation along the river were identified and removed from the DEM. Detection was carried out using longitudinal profiles along the corrected river axes. The procedure used the first and second derivatives of the height profiles in combination with assumptions about the shape and minimum size of the obstacles. All segments of the river axes that significantly interrupted the monotonous elevation decrease were marked, and the obstacles were eliminated from the DEM by cross-section-based interpolation of the river bed (Figure 8). A total of around 42,500

360 bridges or other obstacles were detected across Austria and removed from the DEM. In about 1500 cases an increase in the elevation along the river axis was interpreted and implemented as culverts without changing the DEM, based on the interpretation of the DOP and the DEM by the analyst. On large rivers, information about bridge piers and power plants was often available and this was taken into account in the hydrodynamic modelling, while on small streams the bridges were assumed to be non-existent.

365 To assist in the further analyses, river banks were detected, and two line pairs were added to each centre line of the river axis: the upper edge of embankment to represent the intersection with the flood plain, and the lower edge of embankment as an approximation of the water-land-interface during average flow conditions. Both edges were identified by cross-section-wise analysis of the DEM along the river. First, the centres of the embankments on both sides of the river channel were detected as local transversal slope maxima in proximity of the centre line. Starting from each centre of embankment, the

370 cross section was followed away from the channel and the upper edge of embankment was identified as a minimum of curvature. Similarly, starting from each centre of embankment, the cross section was followed towards the channel and the lower edge of embankment was identified as a maximum of curvature.

As the DEM from LiDAR represents the water surface rather than the river bed, bathymetric information from other sources was integrated: (i) mesh terrain data from existing, local hydraulic models provided by the individual states (16% of the river

375 network), (ii) measured cross-sectional profiles along the river axis (2% of the river network) (Mandlburger, 2000) and (iii) estimates of cross-sectional area assuming a trapezoidal channel geometry and using Manning's equation with longitudinal slope, roughness and discharge measured at stream gauges, as well as water level and width of the channel (82% of the river





network). Between the lower embankment lines, the DEM was replaced by this bathymetric information and the transition was smoothed.

For simulating the case of potential levee failure in the $Q_{300}$ scenario, a topography without levees was necessary. Since no levee data base exists in Austria, they were detected from the DEM. Levee detection was carried out by calculating a moving average of terrain heights with window size $w$ in the $Q_{300}$ floodplain, determining its difference to the original DEM, and interpreting differences larger than a height threshold as objects. The window size was set to $w = 50$ m based on test calculations. The result was a raster difference model. The total length of detected levees was about 3000 km. An example,

of a levee detected by this method is shown in Figure 9.

**5.2 Combination of automatic and manual methods**

Due to the high quality requirements and a myriad of special cases, nearly all analysis steps involved some manual checking and/or correction by an analyst, falling into three categories: (i) Manual checking and modifications within the entire domain was carried out for the position of all river axes. The manual analysis was adopted to enhance quality and since part of the

river network cannot be identified automatically from the DEM because it is underground or obstructed by other objects such as vegetation and buildings. (ii) Spot checks at automatically selected locations were conducted to ensure that the automated methods worked as intended. This approach was adopted for DEM harmonisation, obstacle detection, identification of river banks and levee detection. (iii) Extensive automatisation was carried out for obstacle removal and burning channels into the DEM, as they were well-defined tasks, although some random manual checks were made.

**5.3 Accuracy of the results**

The manually corrected river axes were used to evaluate the quality of the automatic positional correction of the original river network. The original axes had overall 85.0 % of their points within the river channel, as measured against the manually corrected lower embankment lines (where available) which were deemed to be accurate. The automatic correction of the river axes increased this value to 97.1 %. This percentage was lower for small streams than for large streams, mainly

because of the narrower channels.

The automatic detection of obstacles was tested by intersecting the OpenStreetMap traffic route-layer with the river axes in the states of Vorarlberg, Salzburg, Wien and Burgenland, from which 7336 locations of potential bridges were obtained. These were manually checked against the orthophoto. In 4244 cases (57.9 %), the original DEM was correct in that there was either no obstacle (e.g. the crossing was a tunnel and there was no bridge) or it had been eliminated in the original DEM.

In 2970 cases (40.5 %), the automatic algorithm had correctly detected and removed the obstacle. In 122 cases (1.7 %), there was an obstacle in the orthophoto which however had not been identified by the automatic procedure. These errors occurred mainly along small streams and for low bridges with little effect on the height profile of the river in the DEM.





The quality of other analysis steps, such as levee detection, bathymetric data integration and river bank detection, depends on the input data quality. For example, the input data to the bathymetric data integration are the DEM containing the water surface, the bathymetric raster model of the river channel and the lower embankment lines. The estimated trapezoidal channel geometry was tested against measured profiles every 100 river km for a total of 63000 profiles by evaluating the relative area deviation, *RAD*:

$$RAD = \frac{A_m - A_R}{A_r}$$

Eq. 5

where $A_M$ and $A_R$ are the modelled and observed wet cross-sectional areas. The modelled trapezoidal area was calculated using Manning's equation under the assumption of a discharge corresponding to 70 % of the mean annual discharge. Figure 10 shows the evaluation for a reach of the Inn river in Tyrol. The errors are relatively small but do exist. For example, at the location shown in the photo in Figure 10 the wet cross-sectional area is overestimated because of a sandbar associated with the river bend, an effect not represented in the simple bathymetric model used here. Of course, an increased availability of observed cross sections would enhance the accuracy of the bathymetry.

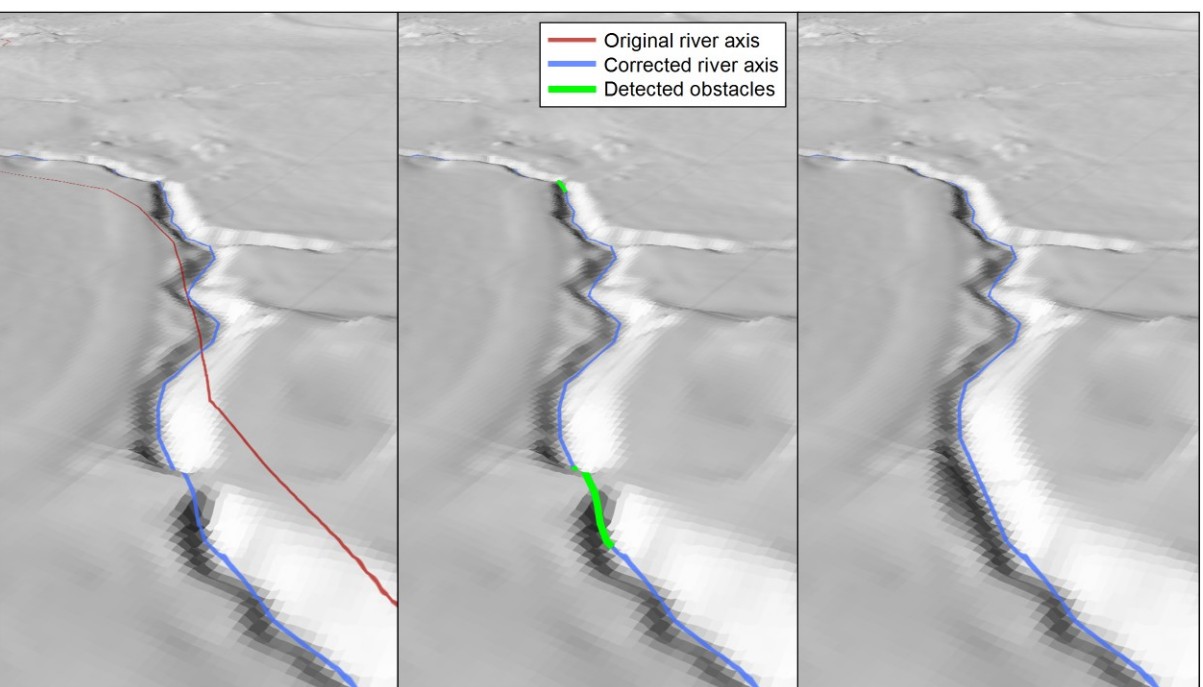

**Figure 8: Example of DEM and river axis corrections. The positional correction of the river axis (left) is followed by obstacle detection (centre) and obstacle elimination (right). The obstacle detected in the foreground is a small bridge. Pixel size is 1 m.**



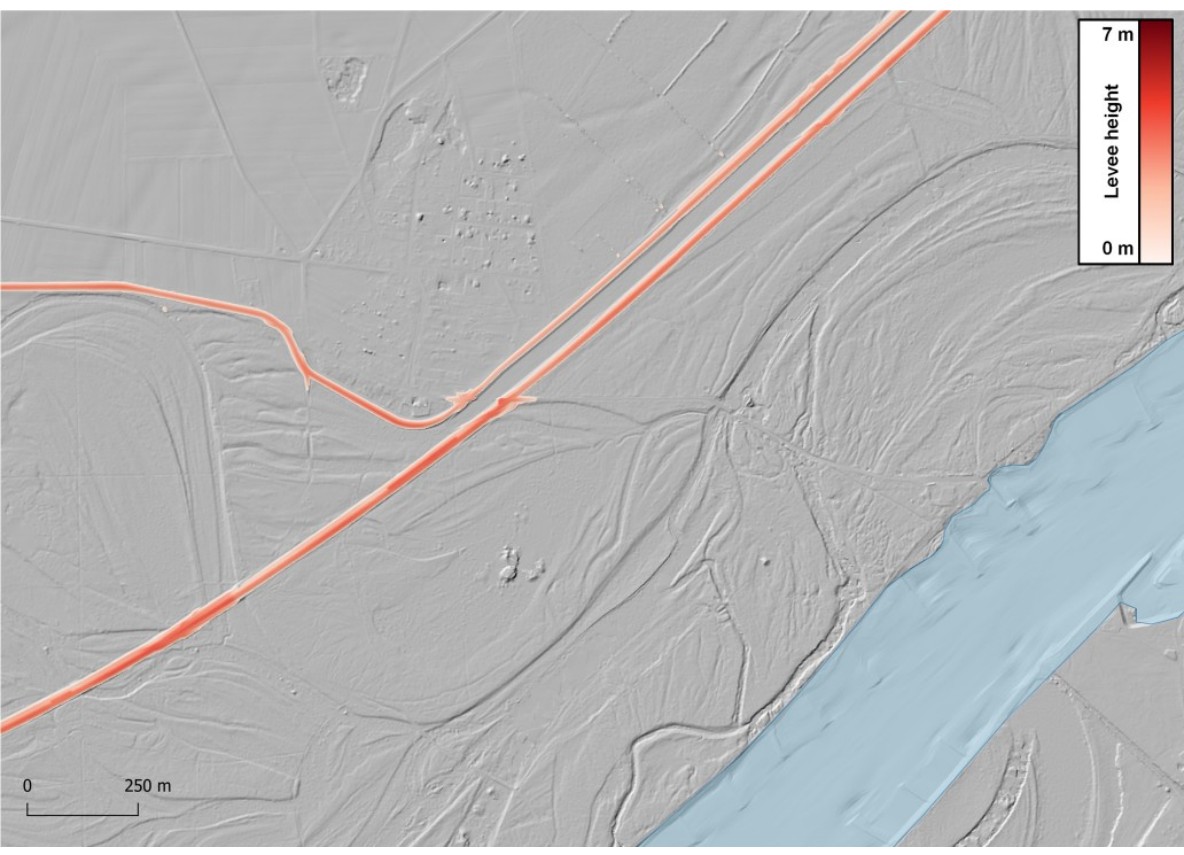


**Figure 9: Levee detection along the Danube near Stopfenreuth, Lower Austria. The two levees with slightly different crest heights protect part of the former flood plains.**


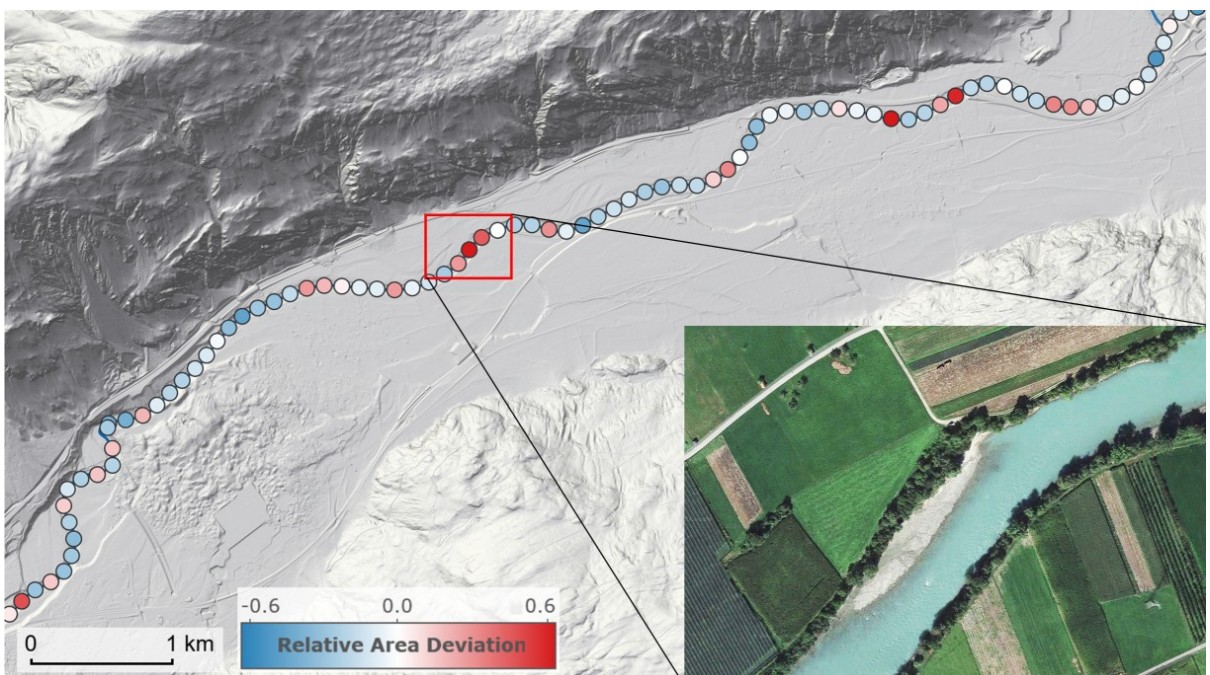

**Figure 10: Validation of the estimated trapezoidal channel geometry on measured profiles of the Inn river near Haiming, Tyrol. Flow from left to right. The relative area deviation, RAD, (shown as filled circles on the river axis) is the relative difference between the estimated and the measured wet cross-sectional area. Red indicates an overestimation of the estimated wet areas, blue an underestimation. Inset photo shows the sandbar that explains the overestimation of the wet cross-sectional area at this location.**

## 435  6. Inundation patterns and velocities

### 6.1 Data and methods used

Transient, two-dimensional hydrodynamic simulations with a spatial resolution of 2 m were carried out in order to calculate water depths and flow velocities. The simulated maximum water depths at every location were taken as relevant for the flood maps.

Manning's roughness coefficients were estimated by combining various pieces of information: On the floodplains, tabulated coefficients from the literature were used based on a landuse classification (Chow, 1959; Arcement and Schneider, 1989). In the stream channels, the roughness coefficients were calibrated against stage-discharge relationships at stream gauges, spatially interpolated and combined with any locally available roughness coefficients from previous studies. The latter were available for 10% of the total river network. This combined approach ensures spatial consistency and exploits local

information where available.


The simulations were not carried out for a specific event, but for peak discharges associated with a return period of $T$=30, 100 and 300 years. The peak discharge and the corresponding water levels therefore represent a hydrological longitudinal section along the stream. This choice differs from the usual procedure of local hydrodynamic modelling, in which an observed event is scaled to a $T$-year peak discharge on the main river, and the flow of the tributaries is scaled in the same
way, which often corresponds to a much smaller return period. In other words, the scenario usually consists of an observed hydrograph multiplied with a constant factor in the entire simulation domain. In contrast, the method chosen here assumes the same return period of peak discharge over a large region, which allows calculation of spatially consistent inundation areas. However, this method is not mass-conserving for two reasons. First, the sum of the $T$-year discharges of two tributaries at a confluence is usually not equal to the $T$-year discharge downstream of the confluence. This is because, in
reality, the peaks of the tributaries rarely occur simultaneously, so there is no perfect superposition of peak discharges. Second, there are diffuse lateral inflows along the river reaches that need to be accounted for. In order to efficiently simulate this effect, water was removed or added at nodes to such an extent that the condition of a spatially uniform return period of the peak discharge (given by the hydrological regionalisation as described in Section 4) was met. At confluences, water was usually removed to compensate for the fact that flood hydrographs do not always occur simultaneously. Water was usually
added along the channel reaches to account for diffuse lateral inflows. The extractions and additions were dynamic, so that not only the peak discharge but the entire specified discharge hydrograph at all nodes (from section 4) was retained (see Figure 11). In this way, the simulated discharge hydrograph at all nodes of the river network were consistent with those obtained by the hydrological regionalisation. Details of the method can be found in Buttinger-Kreuzhuber et al. (2022).

In order to enhance simulation accuracy in urban regions, buildings and culverts were represented explicitly. Culverts were
modelled by standard flow formulas accounting for their diameters, pressure heads, and inlet and outlet geometries. High-head and run-of-river power plants were represented by prescribing a stage-discharge boundary condition and by incorporating the weir geometry in the DEM, respectively. In total, 1475 culverts and 65 high-head power plants were specified semi-automatically.

The 2D shallow water equations were solved by a second-order accurate finite-volume scheme (Buttinger-Kreuzhuber et al.,
2019). The algorithms were parallelised (Horvath et al., 2016) and implemented on ten NVIDIA Titan RTX graphics processors, each with 24 GB of video memory (Buttinger-Kreuzhuber et al., 2022). The GPU implementation was optimised by allocating only those regions that were wet or at risk of getting wet (Horváth et al., 2016). Austria was divided into 182 simulation domains with a total of around 20 billion cells. The time step was constrained by the Courant-Friedrichs-Lewy-condition and was typically less than 1 second. With this implementation, the simulation time for the 100-year flood case
was 28 days. The simulations were carried out for three different flood probabilities ($T = 30, 100, 300$) and for a scenario $T = 300$ without levees. In addition, some of the simulations were repeated for iterative quality control. The solver was integrated into a dataflow of the automation framework Visdom (Waser et al., 2011) which vastly facilitated the work flow.



## 6.2 In what way the combined automatic-manual methods work

In order to enhance the quality of the simulated flood hazard zones, the burned-in river bed, culverts and hydraulic structures, such as power plants, levees and protection walls, were checked in a semi-automatic way. Automatic checks were performed to ensure the completeness and consistency of the input data. Additionally, test simulations with river discharges equal to mean annual discharge over a few hours were launched, to test flow connectivity and the plausible behaviour of weirs and the burned-in river bed. The simulations were visually examined and apparent errors were manually corrected, e.g., by increasing the aperture of weirs through modifying the DEM geometry. Similarly, underpasses below levees were opened or closed, guided by available local information on flood hazard zones. Occasionally, underpasses that appeared closed in the reference dataset had to be opened to allow free outflow from tributaries. Where relevant information was available, thin walls not resolved in the DEM were added manually, if they were deemed to significantly affect the inundation areas.

## 6.3 Accuracy of the results

In a first step the hydrodynamic simulations were compared with the 100-year flood water levels at individual stream gauges. For the gauges along the Inn in Tyrol, there were usually only deviations of a few centimetres. Exceptions are the Jenbach-Rotholz and Brixlegg stream gauges with water levels that were around 20 to 30 cm lower than observed, which can be explained in part by backwater effects of bridges not fully accounted for.

In a second step, the simulations were evaluated against reference flood hazard maps previously available that are the outcome of assessment process at the community level (Schmid et al., 2022). Since these reference maps were obtained in local studies, often based on observed flood inundations, their accuracy can generally be considered higher than that of the present simulations. However, they were only available along 11600 km (34 %) of the river network. Specifying the modelled state of each cell as either wet ($M_1$) or dry ($M_0$) and the state of each cell of the reference as either wet ($R_1$) or dry ($R_0$), the following performance metrics were used:

$$HTR = \frac{|M_1 R_1|}{|R_1|}, \quad FAR = \frac{|M_1 R_0|}{|M_1|}, \quad CSI = \frac{|M_1 R_1|}{|M_1 R_1 + M_0 R_1 + M_1 R_0|} \qquad \text{Eq. 6}$$

where the hitrate $HTR$ reflects the degree to which wet cells are captured; the false alarm rate $FAR$ the relative number of cells that are wet in the model but dry in the reference data; and the critical success index $CSI$ the number of correctly predicted wet cells relative to the total number of wet cells in the model and/or the reference (Buttinger-Kreuzhuber et al., 2022).

These three performance metrics and the area $A_{MR}$, which is the sum of the cell areas that are flooded both in the proposed model and in the reference, are shown for return periods of 30, 100, and 300 years in Table 2 for Tyrol. As the return period increases, the hitrate changes little, the false alarm rate decreases and the critical success index increases. The decreasing false alarm rate reflects the more robust results for greater water depths, and the smaller influence of small streams. This tendency is consistent with other studies (Bates et al., 2021).





Additionally, the patterns of the simulated and reference maps were compared in order to get more detailed process insights of the performance (Grayson et al., 2002). Overall, the patterns show an excellent agreement. For example, the Eferding area along the Danube that was heavily inundated during the 2002 and 2013 floods (Blöschl et al., 2013) shows small differences (Figure 12a). In this case, the accurate representation of the Ottensheim-Wilhering run-of-river power plant and the use of instationary, second-order accurate simulations played a central role. In individual cases, however, there are deviations due

to a number of reasons. First, the present simulations represent a consistent flood return period across the entire river network, while the references maps were obtained by the traditional approach of mass conserving (design event) simulations. The resulting differences mainly occur around confluences. Second, some of the small streams are not included in the local flood hazard zones, which may cause false alarms near confluences. Third, the reference maps sometimes reflect local information not available for the national scale simulations conducted here. These pieces of information include walls with a

width < 2m, and some culverts and mill channels. An example is shown in Figure 12b for illustration. In the original simulations the western part of the community is flooded (blue) while the local maps reflect a protection by a narrow wall in the south of the area. Including a thin wall (Figure 12c), results in simulations much closer to the reference map.

The inundation areas, water depths and associated flow velocities represent the output of the flood hazard mapping. An example is shown in Figure 13 for the Gail valley near Feistritz. The differences between Figures 13a and 13b reflect the

added hazard when moving from flooding with a return period of 30 years to that of 300 years when levees are overtopped or fail.

The flood risk areas determined in this way with a resolution of 2 m were published in map form on the HORA platform (www.hora.gv.at). For those river sections in which community level flood hazard zones were available, these were shown instead of the present simulations.



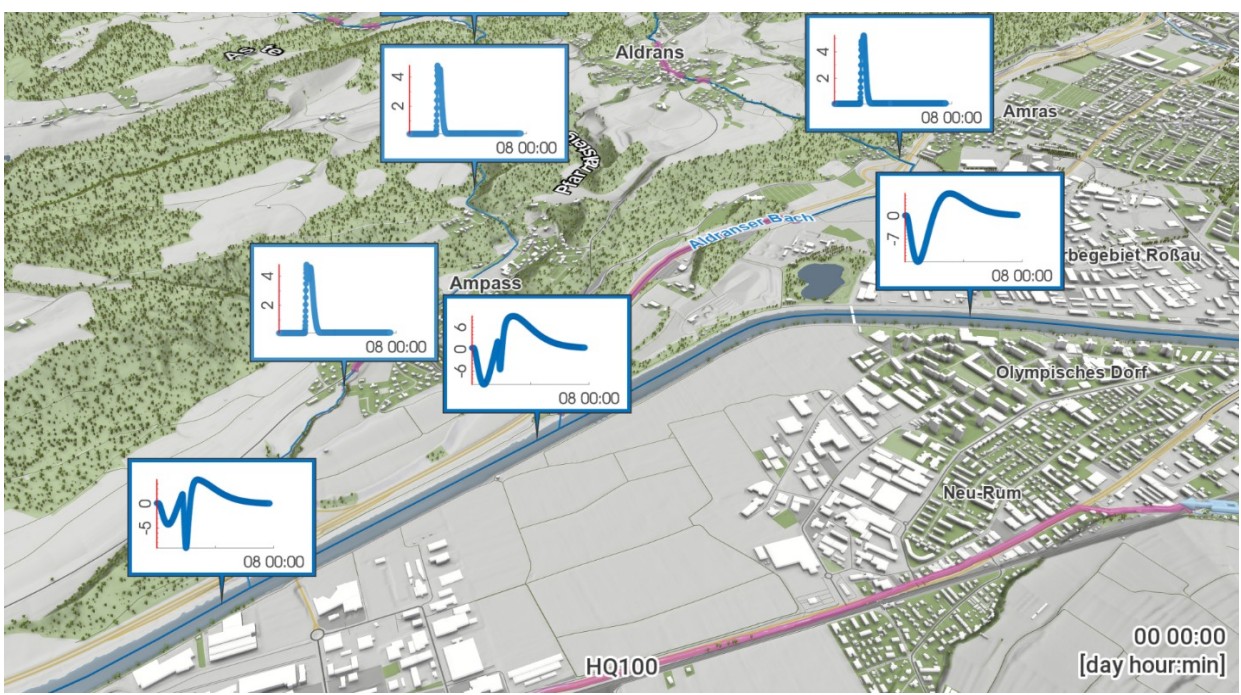

**Figure 11: Mass balance adjustments for the hydrodynamic simulations. The main river is the Inn in Tyrol, flow from right to left. Water is added at the tributaries to reflect lateral inflows and is removed or added at the confluences to reflect the coincidence of flood events in a way to match the discharge hydrographs prescribed by the regional flood frequency analysis. The horizontal axes**
**of the adjustment hydrographs represent time (a total of 8 days), the vertical axes discharge (m³/s).**

**Table 2: Performance metrics of the hazard zone simulations against local flood hazard maps in Tyrol. Three scenarios associated with peak discharge return periods of 30, 100 and 300 years. The metrics are the hit rate (*HTR*), the false alarm ratio (*FAR*) and the critical success index (*CSI*) (Eq. 6). For context, the area $A_{MR}$ that is wet in both the model and the reference is also given. A**
**perfect match of the simulations with the reference data implies *HTR*=1, *FAR*=0, *CSI*=1.**

| Return period | *HTR* | *FAR* | *CSI* | $A_{MR}$ [km²] |
|---|---|---|---|---|
| 30 years | 0.802 | 0.321 | 0.581 | 59.28 |
| 100 years | 0.789 | 0.226 | 0.641 | 94.85 |
| 300 years | 0.802 | 0.155 | 0.699 | 131.33 |



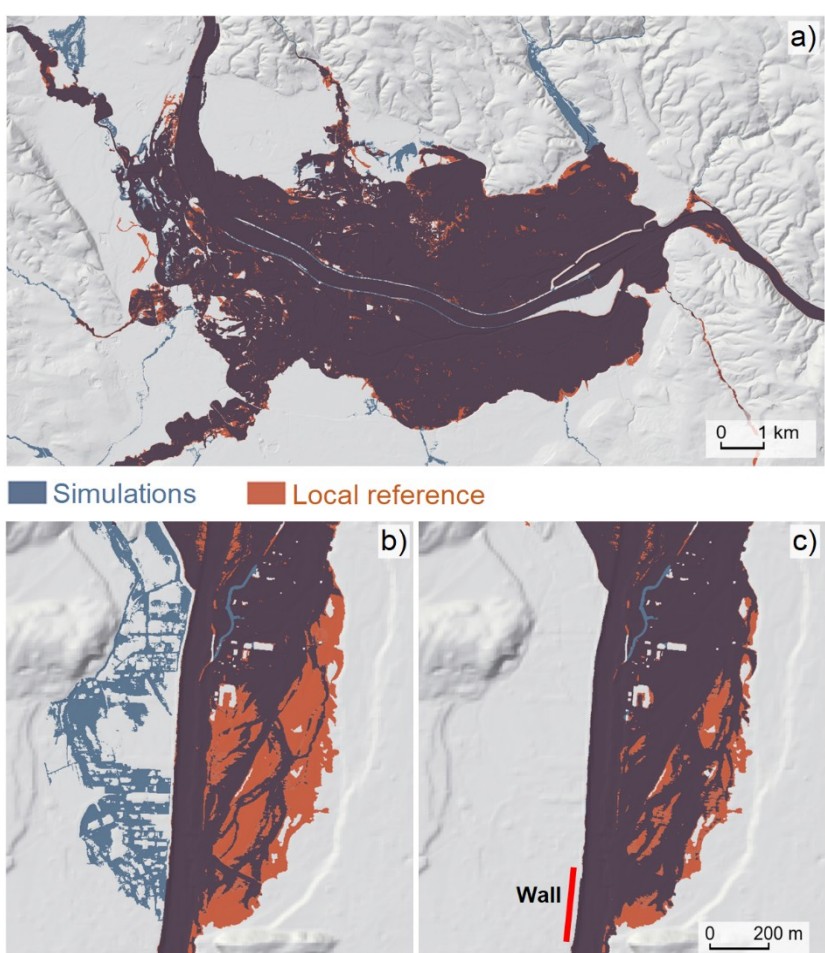

**Figure 12: (a) Comparison of the simulated 100 yr flood hazard zones with local reference flood hazard maps in the Eferding area,**
**Upper Austria along the Danube. There is very good agreement because of the availability of detailed data on hydraulic**
**structures. (b) Similar comparison for Reutte, Tyrol, along the Lech river, for which a protective wall was not included, so**
**simulations overestimate inundation on the left. (c) As in (b) but including the protective wall.**





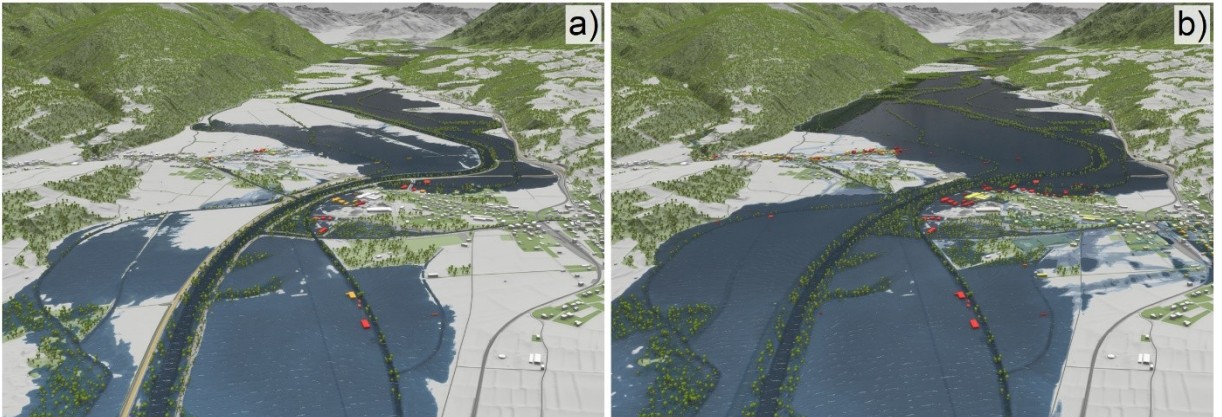

**Figure 13: Simulated flood hazard maps in the Gail valley near Feistritz in Carinthia for (a) a return period of $T$ = 30 years and**
**(b) $T$ = 300 years with levees removed as an indicator of the excess risk (levee breaches).**

## 7. Discussion and conclusions

### 7.1 Spatial flood probabilities

Flood hazard maps are similar to inundation maps in that they show part of the landscape inundated and the rest not, but
conceptually they are quite different. Technically, the flood hazard is defined as the probability with which a given location
is inundated, but in practice one uses peak discharges of a given probability as a boundary condition of the hydrodynamic
models to estimate the inundation areas, assuming that the two probabilities are the same (Mudashiru et al., 2021). For one
cross section this assumption is satisfied if the stage-discharge relationship is unique and monotonic, but spatially this is not
necessarily the case because of dynamic effects (Schumann et al., 2016). The T-year discharge on the other hand can be
estimated from rainfall or directly from flood discharge data. The latter tends to be more reliable if stream gauges are
available in the region (Rogger et al. 2012) because of the complex mapping between rainfall and flood probabilities
(Viglione et al., 2009; Breinl et al., 2021), and this is the reason why this method was adopted in the present study. The use
of the T-year discharges on the entire river network, however, introduces another complication, as these discharges do not
represent a single event. As a consequence, the traditional event-based assumption of local hazard mapping where, say, a
100-year flood on the main stream is combined with 10 year floods on the tributaries to resemble a real event, no longer
applies. One way of addressing this problem is the simulation of streamflow series over hundreds or thousands of years
(either from rainfall or from stochastic discharge models), use of these for hydrodynamic simulations (e.g., Domeneghetti et
al., 2013; Falter et al. 2015, 2016) and performance of extreme value analysis at the end. While conceptually simple, it is not
numerically efficient. This paper therefore proposes an alternative approach, which simulates only a few days of flow



processes, using discharge hydrographs with T-year peaks on all nodes of the river network, but correcting the mass balance of the hydrodynamic model. In the approach adopted here, water is removed or added at each node in a way to obtain a match between the hydrograph simulated by the hydrodynamic model with the one imposed by the regional flood frequency analysis downstream of the confluence. The gain in computational efficiency as compared to the stochastic simulation of long series is a number of orders of magnitude.


**7.2 Scale issues of hyperresolution modelling**

The hyperresolution approach adopted in this study entails a number of scale issues that do not occur in smaller domains or at lower resolutions. For example, in the definition of the river network, the effect of water management structures, such as diversion channels and weirs, on the flow connectivity had to be represented. It is possible to capture the detailed hydraulic

characteristics of these structures for individual cases (e.g. by surveying the geometry and the field and resorting to the operation rules). However, for many such structures this is no longer possible for logistic reasons, so an approximate approach was introduced that was inspired by the large-scale field-mapping method of Reszler et al. (2018) that combines map information (including aerial photographs, local knowledge and expert judgement). In the present case it was not formalised to allow flexibility for different types of hydraulic structures and owing to the fact that no complete national-scale

data base of hydraulic structures exists in Austria. Similarly, in the estimation of flood generation for a T-year discharge peak, such local structures are relevant, in particular flood retention basins. Again, given that operation rules of hydraulic structures were only known in a few cases, approximate approaches were needed based on level-pool routing and standardised hydraulic characteristics. There are simpler alternatives, such as the FARL (flood attenuation by reservoirs and lakes) index of the FEH (1999) and total storage based methods (e.g. Wang et al, 2017), as well as more

complex methods (e.g. Ayalew et al. 2013; Connaughton et al., 2014) but it is clear that the choice of method needs to be aligned with the amount and nature of data at hand. In the definition of the terrain geometry, it was possible to adopt a higher level of automation given the availability of a Lidar DEM and the somewhat more universal nature of the geometric shape of bridges and levees. While methods that are not shape-sensitive exist (e.g. Sithole and Vosselman, 2006) there is an element of subjectivity in the parameter choices made, e.g., the kernel size in the levee detection method used here.

Regarding the hydrodynamic simulations, the most relevant hydraulic structures (e.g. culverts, reservoirs, mobile walls) were included explicitly where known. Again, there is some element of subjectivity in terms of which structures were included related to both data availability and expert judgement in the visual screening of the hydraulic situation of the entire computational domain. As with some of the other parts of the analysis, a decision needs to be made on the state of the hydraulic system in the scenario considered (e.g. mobile walls absent/present, reservoir empty/full, weir open/closed) which

represents meta-information of the flood hazard maps produced.





The scale issues related to local hydraulic structures and water resources operation are quite different from the sub-grid scale parameterisations hydrology, hydraulics and terrain analysis traditionally has dealt with (Blöschl and Sivaplan, 1995; Dottori et al., 2013), so variance-based approaches (as in turbulence) are not an obvious choice in developing a more objective method. Instead, it may be worth adopting a classification approach and test its validity in an intercomparison setting. This classification may merge heterogeneous data types (e.g. aerial photographs, terrain data, engineering data bases) with elements of human judgement to address some of the ambiguities and inconsistencies likely to be encountered in large datasets. While the tradeoff between representable level of detail of each structure and the total number of structures accounted for will likely rest with us for the near future, there is potential in pushing the boundary be explicitly recognizing a conceptual typology of structures relevant in flood hazard mapping, in order to make progress in capturing the summary effects of human impacts on flood processes.

### 7.3 Combination of heterogeneous data, manual interventions and accuracy

More generally, this approach of using all available, relevant data as well as some element of human judgement where needed, proved to be very efficient in all steps of the analysis. Part of the heterogeneity is related to hydraulic structures, but other complexities are equally important. For example, the stream gauges are not uniformly distributed in the study region. In the mountainous areas there tend to be more ungauged catchments than in the lowlands and these differences were accounted for by the flood frequency hydrology approach adopted here that combines the flood discharge record with temporal, spatial, and causal expansion of information (Merz and Blöschl, 2008ab). While more formal Bayesian approaches exist (Viglione et al.2013) we opted for a more informal, albeit more labour intensive, approach to obtain the flexibility to draw from information not contained in the data without having to set up a-priori distributions. Of course, future work might consider such an approach. Another example is the identification of obstacles along the river network such as bridges, which mostly worked very well but the visually inspection showed that in 1.7% of the cases a manual intervention was needed. We believe that this combined approach has significantly increased the accuracy of the results.

There is a general question as to which extent large scale flood hazard mapping can produce results similar to more local studies. In the present study, in each of the analysis steps (Figure 1) we attempted to mimic local estimation procedures to the extent possible. In estimating the 100-year flood discharge for locations at the river network without stream flow measurements (ungauged basins) the results are similar or better than smaller scale studies in the literature. Here, the root mean squared normalised error (RMSNE) was 0.30, 0.29 and 0.06 for catchments <100 km² area, between 100 and 1000 km² and larger than 1000 km², respectively. Out of the 50 studies summarised in Table A9.1 of Blöschl et al. (2013), only four out report RMSNE of 0.30 or less, and none RMSNE of 0.25 or less.

With regard to the inundated areas, the model achieves a critical success index (CSI) of 0.69 and a hit rate of 83 % across Austria for a 100-year flood. In Tyrol, the CSI scores are 0.58, 0.64 and 0.70 for the 30-year, 100-year and 300-year flood. These values are comparable to other large-scale studies (Wing et al., 2017; Bates et al., 2021). Local studies, e.g. Aronica et





al. (2002), report CSI values of 0.70 to 0.85 for best-pick scenarios in an ensemble. Still, comparisons across different regions need to be treated with caution, as floodplain topography and whether the floodplain is developed or not have implications on model performance (Wing et al., 2017).

A related question is whether which source of uncertainty – estimation of flood discharge or hydrodynamic modelling – has a bigger effect on the resulting flood hazard map. While sensitivity analyses would be needed to address this question for the present modelling setup, there are indications in the literature that the flood discharge estimation may be more important. For example, uncertainty analyses in a coastal catchment in Italy suggest that flood discharge estimation introduces more uncertainty than the hydrodynamic modelling (Annis et al., 2020). However, one can safely assume that the relative importance of this two uncertainty factors will be strongly controlled by the terrain configuration. Flat flood plains will tend to make hydrodynamic processes more relevant, while steeper terrain will make them less relevant, as a given discharge change translates into a smaller change in inundation area.

**7.4 Limitations, potential for improvement and outlook**

Similar to other studies, there is potential of improvements in all steps of the analysis which fall into two categories: ingestion of additional data and refined methods. Regarding the river network and catchment boundaries we believe we have achieved a high level of accuracy, but it would be possible to relax the assumption of a strict tree structure. Each node has currently only one downstream neighbour, but it would be possible to allow for bifurcations, in which case an allocation of the flood flows would have to be made to two (or more) downstream branches. Regarding the flood discharge peaks and volumes, the current setup involves a flood frequency analysis based on observed stream discharges. While we do not believe that at this point (and probably for years to come) deriving flood discharges from precipitation will give more accurate results in the study region because of the availability of flood discharge data, there may be other reasons for doing this, in particular when interested in changing flood hazard as a result of changing climate change and changing land use. The existing stream gauges could still be used for calibrating the runoff model for such an analysis. Regarding the DEM variants and river boundaries, possible extensions are more detailed visual checks to account for even more local information. Another possibility is the processing of the DEM directly based on point clouds which would however increase the computational burden. Regarding the hydrodynamic simulations of the inundation patterns and velocities, a larger number of local hydraulic structures could be ingested, which would require a major digitalisation effort. Another interesting extension is the use of the same system for estimated pluvial flood hazards at the national scale at a resolution of 2 m. To this end, one would start from extreme rainfall and simulate infiltration and runoff generation directly at the pixel scale (Buttinger-Kreuzhuber et al., 2022). The obvious challenge here is capturing the probability of surface runoff during floods (Blöschl, 2022a), although the existing stream gauges could be used for calibration and validation.

The present framework was, as is often the case, developed in the context of an externally funded project with a finite project duration. It is planned that the data base and model codes are maintained at least for a number of years and are updated
where possible. From a project funding perspective an ongoing maintenance is not always easy because of limited budget horizons, but there may be very valuable pay backs through opportunities for future adaptations of the flood hazard maps regarding changes in climate, land use and hydraulic structures in the system (Blöschl et al., 2019; Blöschl, 2022b). A follow-up project developed visualisations of the hazard maps made publicly available on the https://hora.gv.at platform. The visualisation involves personalised 3D perspective views of the inundation of buildings for the hazard scenarios (Cornel et

al., 2015).

The framework clearly lends itself to application in other regions, but there may be additional particularities that need to be accounted for, such as tidal effects in coastal areas. The manual steps are amenable to more formal quantifications, e.g. by classification methods, which could shift the balance of automatic-manual methods more towards automatic ones. These may lead to interaction-aware models involving large-scale statistical dependence models for extremes and high-resolution

climate models coupled to hydrological catchment models providing spatially and temporally consistent meteorological and hydrological hazard estimates (Vorogushyn et al., 2018).

**Code availability**

The Top-kriging code for regionalising the flood data is freely available from https://r-forge.r-project.org/projects/rtop. The

680 Visdom framework used for simulating the inundation patterns and velocities is closed source. Licensing information is available on request from VRVis (https://www.vrvis.at/visdom).

**Data availability**

Flood peak data and discharge data are freely available from https://ehyd.gv.at. River network, catchment boundary and

685 DEM data are owned by the Federal Ministry of Agriculture, Forestry, Regions and Water Management, and can be made available in line with legal requirements. The simulated inundation data may be shared upon request and are displayed at https://www.hora.gv.at. The local flood hazard maps used as reference data set in Tyrol are available on https://data-tiris.opendata.arcgis.com/datasets/ueberflutungsflaechen-1.

**Author contributions**

GB designed and led the study, and drafted the manuscript. JP, PV and RV processed and aligned the catchment and river network datasets. PV regionalized the flood peaks to the ungauged catchments. JP, PV and RV performed the cross-validation of discharges to assess method performance. AB, AR, JE, MW, JP, MHofer and PV performed the data analysis.
JK, JS, JW, MHollaus and NP helped with the data analysis and interpretation. AB, JK, JW, MW and PV helped with developing the manuscript and the figures. AB, AK, DC, JP, JK, JW, MW, PV and ZH implemented algorithms for pre-processing river network, DEM, river bed, land use and roughness data. AB, AK, DC, JW and ZH and implemented algorithms for inundation data post-processing. AB, JW and ZH implemented the numerical inundation model. AB performed the validation of the inundated areas against the local reference flood maps. HS provided strategic guidance on the project. All authors gave suggestions on the methodology and contributed to framing and revising the paper.

**Competing interests**

The contact author has declared that neither they nor their co-authors have any competing interests.

**Disclaimer**

Publisher's note: Copernicus Publications remains neutral with regard to jurisdictional claims in published maps and institutional affiliations.

**Acknowledgements**

**Financial support**

This research has been supported by the Austrian Federal Ministry of Agriculture, Regions and Tourism and the FWF Vienna Doctoral Programme on Water Resource Systems (DK W1219-N28). VRVis is funded by BMK, BMDW, Styria, SFG, Tyrol and Vienna Business Agency in the scope of COMET – Competence Centers for Excellent Technologies (879730) which is managed by FFG.

**Review statement**

This paper was edited by xx and reviewed by xx anonymous referees.



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
