# Peer review of "Hyper-resolution flood hazard mapping at the national scale"

_Natural Hazards and Earth System Sciences, 2023_

## Author Response (AR1)

**RC1: 'Comment on nhess-2023-209', Alberto Montanari, 09 Jan 2024**

The paper addresses a relevant technical problem in flood prediction and management. It is well written and optimally organised, it is an enjoyable and instructive reading. I think it is particularly interesting the identification of the three main challenges associated to large scale flood mapping. In particular, I fully agree that representing local details and adopting the same return period for large regions are open and technically relevant challenges.

The study looks very rigorous and innovative in the technical solutions that are presented. The results are rigorously evaluated and the quality of the obtained maps is outstanding. For the originality of the approach, I think it may become a pilot for application in other contexts. Therefore, I am in favour of publication, after minor revisions to address the comments that I am listing below.

Minor comments

A general comment is related to the specificity of the approach. Austria is rich of data and information, so that the combined automatic-manual methods that are suggested here may result very specific. Would the proposed procedure work in contexts where the information is more sparse? What about application to larger areas?

Lines 453-460: the technique that has been used to ensure mass conservation is the most interesting solution that is presented in the paper. I admit that the technique is not clear to me. At line 460 is stated that "The extractions and additions were dynamic". Does this mean that water is continuously added along any river reach? At confluences water is removed abruptly? If yes, is the simulation split into parts, to avoid inconsistencies of the flood inundation maps at confluences? As far as I understand, there might be a discontinuity in the inundated areas if water is artificially removed at a given node. I am sure this issue has been carefully evaluated, so I suggest to expand the explanation of the adopted solution.

Line 150: does the removal of channels introduce an approximation? I understand why channels are removed. On the other hand, the reality include the bifurcations that are artificially removed in the model. What are the authors' recommendations for the application of such procedure in other contexts? What approximation should we expect? This issue is discussed at lines 645-650, but still I think some additional information would be useful.

Line 360: a similar comment. If bridges were removed, how local singularities are taken into account?

Line 106: please complete the reference Rogger et al.

Titles of several sections are identical. I understand that similar procedures are applied to process several sources of information, so using identical titles may be justified. However, I found it difficult to get oriented into the manuscript during my first reading.

Congratulations on an interesting contribution!

Alberto Montanari

**Reply**

We would like to thank the reviewer for the careful assessment of the manuscript. Our replies (R) to the comments (C) are as follows:

C: A general comment is related to the specificity of the approach. Austria is rich of data and information, so that the combined automatic-manual methods that are suggested here may result very specific. Would the proposed procedure work in contexts where the information is more sparse? What about application to larger areas?

R: Yes, Austria is rich of data and information. There are a number of countries that are similar in this respect, and there are perhaps a larger number of countries where data are more scarce. We believe that, in more data scarce situations, the combined automatic-manual approach would still work very well. The difference is that, for both the automatic and the manual steps, less information is available, so uncertainties are larger, but the complementarity of the approaches still exists and contributes to an overall strengthening of the outcomes. We have included a remark to this effect in Section 7.4 of the revised paper to address this important aspect.

C: Lines 453-460: the technique that has been used to ensure mass conservation is the most interesting solution that is presented in the paper. I admit that the technique is not clear to me. At line 460 is stated that "The extractions and additions were dynamic". Does this mean that water is continuously added along any river reach? At confluences water is removed abruptly? If yes, is the simulation split into parts, to avoid inconsistencies of the flood inundation maps at confluences? As far as I understand, there might be a discontinuity in the inundated areas if water is artificially removed at a given node. I am sure this issue has been carefully evaluated, so I suggest to expand the explanation of the adopted solution.

R: Yes, the extractions and additions are time series, and they are adjusted iteratively in a way to obtain consistency between the hydrodynamic model and the prescribed discharge hydrographs at all nodes. The simulation are not split into parts, instead the extractions and additions are sinks and sources of the St. Venant equations. The sinks and sources are spread over an area in the stream in order to avoid any discontinuities. We have added a more detailed explanation in Section 6.1 of the revised paper for clarification.

C: Line 150: does the removal of channels introduce an approximation? I understand why channels are removed. On the other hand, the reality includes the bifurcations that are artificially removed in the model. What are the authors' recommendations for the application of such procedure in other contexts? What approximation should we expect? This issue is discussed at lines 645-650, but still I think some additional information would be useful.

R: Yes, the removal of channels introduces an approximation. Bifurcations, often, are very important for low flow conditions and they lose in importance as discharge increases and water spills beyond the river banks into the flood plains. Again, this is an important point, and we have accordingly included a comment on the effects to be expected of this assumption on the simulated flood inundation areas in Section 7.4.

C: Line 360: a similar comment. If bridges were removed, how local singularities are taken into account?

R: Bridges were removed from the terrain model, so no singularities are introduced. It is true that this is an approximation, as only the backwater effects from the bridge piers are considered, but not those from the bridge decks, as the latter have been removed. A comment has been added to Section 5.1 of the manuscript.

C: Line 106: please complete the reference Rogger et al.

R: Has been completed.

C: Titles of several sections are identical. I understand that similar procedures are applied to process several sources of information, so using identical titles may be justified. However, I found it difficult to get oriented into the manuscript during my first reading.

R: We have revised the titles to enhance the clarity of the layout of the paper.

C: Congratulations on an interesting contribution!

R: Thank you.

**RC2: 'Comment on nhess-2023-209', Félix Francés, 16 Jan 2024**

First of all, I would like to highlight the huge effort behind this project and with an excellent final result. Congratulations!

This paper is clearly oriented to the practical resolution of a problem of enormous importance, such as the elaboration of flood hazard maps at a regional scale with an acceptable accuracy for the "client". From this point of view, I found the optimal combination of manual and automatic procedures to be very good. I was also very pleased with the consistency in the estimation in all steps of the resulting reliability.

I have enjoyed reading this paper, even if it is technical and lacks new scientific contributions. Proof of this is I had many comments and questions while reading it. I have selected and summarized them in the next paragraphs.

HYDROLOGICAL SCENARIOS

The general methodology is clearly presented in chapter 2, but I miss a better State of the Art in the introduction chapter. In particular, I am missing a review of the different methodologies for the hydrological scenarios to be simulated by the hydraulic model. Simplifying: on one hand the methodologies based on hydrological modelling with a design storm, synthetic storms or a weather generator as input; on the other hand, using historical floods or a design hydrograph (as in this paper) without hydrological modelling.

I totally agree that a challenge at regional scale is to adequately extend or assign the flood hydrographs in space (L74-82). Your approach here is based on Buttinger-Kreuzhuber et al. (2022), but why not use synthetic storms or a weather generator? This issue can be discussed in the Introduction and/or in chapter 7.

L650-652. I do not agree: it depends clearly on the flowgauge stations availability (number and record length). In my opinion, using both approaches simultaneously will get the better from both worlds. Can you discuss it here and/or in chapter 7?

SPATIAL RESOLUTION

At large-scale problems, spatial and temporal discretization is crucial from the computational time point of view. At the same time, this discretization may be different for hydrolgy and hydraulics. Some related questions:

The pixel size in this work clearly is 2 m for the hydraulic model. Is it the same for the DEM corrections in chapter 3? Or more general, Are DEM corrections concerning the river network different in chapter 5 for hydraulics than in chapter 3 for hydrology? Yes or not, it will be important to explain why in the new manuscript.

For hydraulic modelling, Why is fixed instead of variable? Numerical problem? In my experience, it can be coarser than 2 m in open areas and, as you explain perfectly, for river channels 2D modeling must be as fine as possible. Why not 1D-2D models for these type of large-scale problems?

RELIABILITY OF REGIONAL FLOOD PEAK ESTIMATION

Regional flood peak estimation usually have a significant uncertainty, mainly because it is difficult to properly consider the land cover and soil characteristics of each catchment. Why not use them in your Top-krigging?

The resulting uncertainty using validation flowgauge stations is presented in section 4.3, but Q100 may be is too big for short lengths. Does results change using Q30?

Table 1 is summarizing the performance and figure 7 is only one example. A scatter plot of observed and interpolated quantiles can be also illustrative to visualize the bias and dispersion.

RELIABILITY OF HYDRAULIC MODELLING

In Section 5.3, it would be interesting to have statistics of RAD for the 63000 profiles, and not only the example of Figure 10. A histogram? Same in Section 6.3 for the metrics described in Equation 6.

OTHER MINOR/FORMAL COMMENTS

L38. Is this traditional or just a rough first approach?

L57. Is this reference just an example for Austria or general? Explain

L62. Is "large scale" an error? Scale is of the problem or resolution? Personally, when reading and to avoid errors, I prefer fine (or finer) scale than large scale. The same: coarse better than small. Or may be scale versus resolution

L97. What do you mean by "collaborators"?

L106. The year is incomplete

L114. More than fourth piece of information is the desired result.

L127. Is billion one thousand of millions (American) or one million of millions (European)? If former, better 20,000 millions.

L32 and L152. The length is a fractal (scaling) property. Is it for the 2 m pixel size? Better, add the total catchment and inundation areas.

L175. Methods or method? May be a short explanation is needed in chapter 2.

L219. I miss that length is for 2019

L244 and others. Equations (numbered as in L244 and not numbered as in L309) are duplicated in the pdf document

L267. I suppose Tc is time of concentration, but better describe it the first time.

L 307. Format of this second level title is different

Table in page 15 can be "nicer"

L464. Culverts are not only present in urban areas. In fact, they are mentioned in general in L361.

L472. Explain the hydraulic modelling concept of "wet": pure hydrologist can confuse it!

Fig 12. May be better with three categories (and colors): agreement, only simulated (overestimation), only local reference (underestimation).

L589-590. Problem with the format.

L599. What do you mean by "mobile walls"? It appears here for the first time.

L601-604. Sorry, I don't understand.

L645. Always in research!

Section 7.4 has a high potential, but it seems a little poor. Can authors be more ambitious? For example, it can be improved from a research point of view.

**Reply**

We would like to thank the reviewer for the careful assessment of the manuscript. Our replies (R) to the comments (C) are as follows:

HYDROLOGICAL SCENARIOS

C: The general methodology is clearly presented in chapter 2, but I miss a better State of the Art in the introduction chapter. In particular, I am missing a review of the different methodologies for the hydrological scenarios to be simulated by the hydraulic model. Simplifying: on one hand the methodologies based on hydrological modelling with a design storm, synthetic storms or a weather generator as input; on the other hand, using historical floods or a design hydrograph (as in this paper) without hydrological modelling.

R: Yes, we used flood observations instead of rainfall-runoff modelling to ensure unbiased estimates of the T-year flood discharges, which are not always easy to achieve by rainfall-runoff modelling. In Section 2 we have now included a review of the alternative methods, albeit brief, as they are not they are not main focus of the paper.

C: I totally agree that a challenge at regional scale is to adequately extend or assign the flood hydrographs in space (L74-82). Your approach here is based on Buttinger-Kreuzhuber et al. (2022), but why not use synthetic storms or a weather generator? This issue can be discussed in the Introduction and/or in chapter 7.

R: Multiple synthetic storms based on a weather generator are indeed an alternative to the approach we adopted and have been used in the past (e.g., Domeneghetti et al., 2013; Falter et al. 2015, 2016, for references see manuscript). While conceptually simple, this approach is not numerically efficient. We say this in Section 7.1 and have now emphasised the advantages of the approach adopted here.

C: L650-652. I do not agree: it depends clearly on the flowgauge stations availability (number and record length). In my opinion, using both approaches simultaneously will get the better from both worlds. Can you discuss it here and/or in chapter 7?

R: L650-652: "… volumes, the current setup involves a flood frequency analysis based on observed stream discharges. While we do not believe that at this point (and probably for years to come) deriving flood discharges from precipitation will give more accurate results in the study region because of the availability of flood discharge data, there may be other reasons for doing … ". Well, in the study region, as we are saying, it is not likely that the rainfall-runoff approach will give more accurate results of the T-year flood than the analysis of flood observations together with expanded information in the sense of Merz and Blöschl (2008a), as our experience from extensive rainfall-runoff modelling in Austria in the last decades has shown. We agree, however, that this will not be the case everywhere, in particular, where runoff observations are scarce. We are now stating explicitly this important point in Section 7.4.

SPATIAL RESOLUTION

C: At large-scale problems, spatial and temporal discretization is crucial from the computational time point of view. At the same time, this discretization may be different for hydrolgy and hydraulics. Some related questions: The pixel size in this work clearly is 2 m for the hydraulic model. Is it the same for the DEM corrections in chapter 3? Or more general, Are DEM corrections concerning the river network different in chapter 5 for hydraulics than in chapter 3 for hydrology? Yes or not, it will be important to explain why in the new manuscript.

R: Yes, The digital elevation model (DEM) is based on LiDAR flights from 2006-2014 with a spatial resolution of 1 m. Additionally, a digital surface model (1 m) derived from the LiDAR

data and a digital orthophoto (DOP, 2013-2015, 0.2 m) including meta information were used. This is stated at the beginning of Section 5.1. Chapter 3 is mainly based on vector data.

C: For hydraulic modelling, Why is fixed instead of variable? Numerical problem? In my experience, it can be coarser than 2 m in open areas and, as you explain perfectly, for river channels 2D modeling must be as fine as possible. Why not 1D-2D models for these type of large-scale problems?

R: We tested various model setups, and it turned out that a fixed grid size is numerically most efficient for the implementation used. This is because the benefit of using a coarser grids in flat parts of the landscape is offset by the additional data management needed to organise and processes unequal grid sizes. With a resolution of 2 m, combined 1D-2D models are not really needed as the streams can be resolved, given that the interest of the present study was on streams draining catchments greater 10 km². We have added a comment to address this point in Section 6.1.

**RELIABILITY OF REGIONAL FLOOD PEAK ESTIMATION**

C: Regional flood peak estimation usually have a significant uncertainty, mainly because it is difficult to properly consider the land cover and soil characteristics of each catchment. Why not use them in your Top-krigging?

R: Top-kriging does account for these uncertainties, as errors associated with each observation are used when solving the kriging system (Skøien el al., 2006). Top-kriging will therefore not only give an estimate of the expected value but also an estimate of its uncertainty. We now explain the treatment of uncertainty in the Section 4.1.

C: The resulting uncertainty using validation flowgauge stations is presented in section 4.3, but Q100 may be is too big for short lengths. Does results change using Q30?

R: The estimation of Q100 is not simply based on flood frequency statistics, but on flood frequency hydrology, in which additional information is used to improve the estimation accuracy (Merz and Blöschl, 2008a, 2008b; Viglione et al., 2013). We also tested the estimation accuracy of Q30 and it turns out that the differences of the uncertainties of Q30 and Q100 are small.

C: Table 1 is summarizing the performance and figure 7 is only one example. A scatter plot of observed and interpolated quantiles can be also illustrative to visualize the bias and dispersion.

R: We have added a new figure with scatter plots of the quantiles to Section 4.3 to address this point.

**RELIABILITY OF HYDRAULIC MODELLING**

C: In Section 5.3, it would be interesting to have statistics of RAD for the 63000 profiles, and not only the example of Figure 10. A histogram? Same in Section 6.3 for the metrics described in Equation 6.

R: We included examples here as, we think, they shed more light on the reasons for the differences between estimates and observations. While we have included a new figure on the

performance of the flood estimation in Section 4.3, we believe that here an additional figure is not needed.

   OTHER MINOR/FORMAL COMMENTS

C: L38. Is this traditional or just a rough first approach?

R: In Austria, during most of the 20th century until the 1970s, the mapping was based on local experience as models were not available, and this had the advantage that local information was accounted for.

C: L57. Is this reference just an example for Austria or general? Explain

R: This is a general overview article.

C: L62. Is "large scale" an error? Scale is of the problem or resolution? Personally, when reading and to avoid errors, I prefer fine (or finer) scale than large scale. The same: coarse better than small. Or may be scale versus resolution

R: Large scales in the sense of Blöschl and Sivapalan (1995). We have added a reference for clarification.

C: L97. What do you mean by "collaborators"?

R: We refer to the collaborators of the study, which we have completed in the text.

C: L106. The year is incomplete

R: Completed

C: L114. More than fourth piece of information is the desired result.

R: Yes, we have added this.

C: L127. Is billion one thousand of millions (American) or one million of millions (European)? If former, better 20,000 millions.

R: A billion is 10^9, which we added for clarification.

C: L32 and L152. The length is a fractal (scaling) property. Is it for the 2 m pixel size? Better, add the total catchment and inundation areas.

R: The lengths are not on a 2m grid, but the length of the vector data. The total catchment area is the entire size of Austria, and less meaningful in this context.

C: L175. Methods or method? May be a short explanation is needed in chapter 2.

R: We believe it is a set of methods, so plural is appropriate.

C: L219. I miss that length is for 2019

R: Last year of record is mentioned at the beginning of Section 4.1

C: L244 and others. Equations (numbered as in L244 and not numbered as in L309) are duplicated in the pdf document

R: Yes, there is indeed a problem with the compiler (correct in the word file)

C: L267. I suppose Tc is time of concentration, but better describe it the first time.

R: Tc is a time scale described at the beginning of the paragraph.

C: L 307. Format of this second level title is different

R: Changed to bold

C: Table in page 15 can be "nicer"

R: Table centered to make it look nicer

C: L464. Culverts are not only present in urban areas. In fact, they are mentioned in general in L361.

R: We removed "urban regions" to make the statement more general.

C: L472. Explain the hydraulic modelling concept of "wet": pure hydrologist can confuse it!

R: We now explain the concept.

C: Fig 12. May be better with three categories (and colors): agreement, only simulated (overestimation), only local reference (underestimation).

R: We have modified the figure to show three colours in the legend.

C: L589-590. Problem with the format.

R: Corrected

C: L599. What do you mean by "mobile walls"? It appears here for the first time.

R: Mobile flood protection walls, corrected.

C: L601-604. Sorry, I don't understand.

R: We have added a comment to explain.

C: L645. Always in research!

R: Yes.

C: Section 7.4 has a high potential, but it seems a little poor. Can authors be more ambitious? For example, it can be improved from a research point of view.

R: Section 7.4 is already rather long relative to other parts of the paper, so we prefer to keep it as is.